# The Active Faults of Eurasia Database (AFEAD): Ontology and Design behind the Continental-Scale Dataset

Egor Zelenin[1], Dmitry Bachmanov[1], Sofya Garipova[1], Vladimir Trifonov[1], Andrey Kozhurin[2, 1]

[1]Geological Institute, Moscow, 119017, Russia

[2]Institute of Volcanology and Seismology, Petropavlovsk-Kamchatsky, 683006, Russia

*Correspondence to*: Egor Zelenin (egor.zelenin@ginras.ru)

**Abstract.**

Active faults are those faults on which movement is possible in the future. It draws particular attention to active faults in geodynamic studies and seismic hazard assessment. Here we present a high-detail continental-scale geodatabase: The Active

Faults of Eurasia Database (AFEAD). It comprises 48,205 objects stored in the shapefile format with spatial detail sufficient for a map of scale 1:1M. Fault sense, a rank of confidence in activity, a rank of slip rate, and a reference to source publications are provided for each database entry. Where possible, it is supplemented with a fault name, fault zone name, abbreviated fault parameters (e.g., slip rate, age of the last motion, total offset), and text information from the sources. The database was collected from 612 published sources, including regional maps, databases, and research papers.

AFEAD facilitates a spatial search for local studies. It provides sufficient detail for planning a study of a particular fault system and guides deeper bibliographical investigations, if needed. This scenario is particularly significant for vast Central and North Asia areas, where most studies are available only in Russian and hardcopy. Moreover, the database model provides the basis for GIS-based regional and continental-scale integrative studies.

The database is available at https://doi.org/10.13140/RG.2.2.25509.58084 (Bachmanov et al., 2022) and via web map at

http://neotec.ginras.ru/index/mapbox/database_map.html (last access May 05, 2022). Database representations and supplementary data are hosted at http://neotec.ginras.ru/index/english/database_eng.html.

## 1 Introduction

The concept of an active fault emerged to distinguish a specific group of faults with present tectonic movements and hence with anticipated activity in the nearest future. The term "active fault" and its synonym "living fault" were introduced in the

late 1940s to 1950s by both American and European authors (Wallace, 1949; topic issue of Geologische Rundschau, 1955). This group of faults has a particular significance in two aspects of geological studies. The slip at the fault produces an earthquake; therefore, active faults are a crucial component of seismic hazard assessment (e.g., Ulomov et al., 1993; Basili et al., 2013; Walker et al., 2021). Moreover, active faulting occurs simultaneously across the Earth and thus provides a basis for studies of recent geodynamics (e.g., Rukieh et al., 2005; Schellart and Lister, 2005; Kozhurin and Zelenin, 2017).

The first global-scale inventory of active faults was the Project II-2 World Map of Major Active Faults of the International Lithosphere Program (ILP) initiated in 1989 and included in the ILP Global Seismic Hazard Assessment Program in 1993. The Project II-2 joined more than 70 scientists from 50 countries lead by two co-chairmen representatives for the Eastern (V.G. Trifonov) and Western (M.S. Machette) Hemispheres. The Geological Institute of the Russian Academy of Sciences, Moscow, hosted the data on active faults of Eurasia provided by project members. These source data were obtained in different scales

and formats (maps, tables, descriptions, and papers), so the first database (DB96) of active faults of Eurasia (Ioffe et al., 1993; Ioffe and Kozhurin, 1996; Trifonov, 1997) was intended to store digitized fault geometry in uniform scale of 1:5M and with a unified set of attributes. Recent advances in tectonics and IT highlighted the limitations of DB96: that outdated database scheme became incompatible with modern GIS, the fault locations lacked accuracy, and many recently studied faults were to be incorporated.

All these issues required creating a conceptually new database, and this work was initiated based on DB96 some 15 years ago (Bachmanov et al., 2017). The authors have designed a new database and GIS for data processing that inherited all the strengths of DB96 but provided far more opportunities for parameterization and analysis. The result of this work is the Active Faults of Eurasia Database (AFEAD), presented in this paper (Fig. 1) and distributed as a shapefile (Bachmanov et al., 2022). In the last decade, active fault databases have been published for some countries (e.g., Atanackov et al., 2021; Ganas, 2021; Jomard et

al., 2017) and subcontinental regions (e.g., Middle East: Danciu et al., 2018; Central Asia: Mohadjer et al., 2016). A global collection of active faults was compiled within the Global Earthquake Model project (Christophersen et al., 2015; Styron and Pagani, 2020). However, none of the published datasets covers the entire Eurasia with uniform detail, and a great lack of data exists for areas of the former USSR. AFEAD addresses this lack as it includes a number of sources published in Russian or hardcopy only or both; ca. 350 publications in total.

AFEAD provides an active fault pattern for the continent of Eurasia and its adjacent water bodies north of 20°S and within 30°W – 180°E. Spatial detail of AFEAD is uniform and equal to 1:1M hardcopy map across the entire dataset to keep a balance between a large amount of data for well-studied regions (such as the Mediterranean) and few pieces of data for least studied areas (such as NE Siberia). There may be a significant time lag between the release of recent publications and their inclusion into AFEAD, and the reader is advised to support AFEAD data with a query to academic databases.

**2 The Concept of an Active Fault**

    Each object of the database represents an active fault. The meaning of this term varies significantly among studies; therefore, we consider it crucial to discuss what kind of data comprise the AFEAD.

    From the most general approach, active faults are those faults on which movement is possible in the future (see discussion in Slemmons and DePolo, 1986). Movements at the fault typically are intermittent, with strong earthquakes and long quiescence

between them. The repose period generally is much longer than human's life, so that sole present-day observations cannot resolve uncertainty in fault activity, making it necessary to study the geological history of the fault.

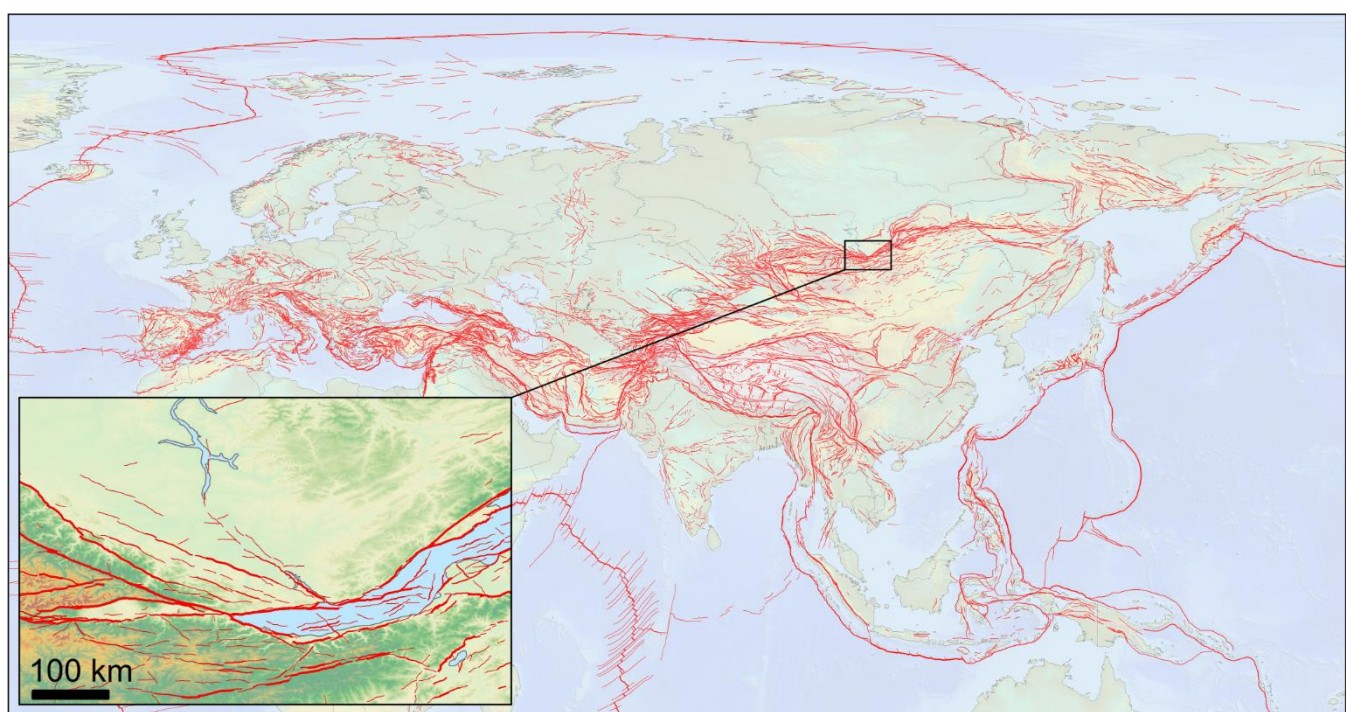

**Figure 1: Overview and detail of the Active Faults of Eurasia Database (AFEAD).**

The expectancy of future movement is what makes a fault active. All other fault parameters (such as kinematics, an average
rate of movement, or any other) are intrinsic to faults in general, disregarding their age, and cannot affect the problem of
activity. The crucial question is what evidence could provide the basis for expecting future movements, and this question
traditionally has been solved via the determination of a "critical" timespan back from the present during which at least one
fault movement could have occurred. It is assumed that if such a movement has occurred, then the fault should be considered
active. Estimations of the "critical" period were summarized by Galadini et al. (2012), and all the cited intervals fall within the
range of 10,000 to 1,000,000 years. However, even longer intervals were discussed (e.g., 2.6 Ma by Atanackov et al., 2021).
The concept of the critical interval implies that a slip may occur at an active fault after long quiescence. Paleoseismological
studies, however, provide recurrence periods of hundreds to the first thousand years with maximum values of order tens ka.
For example, Umehara Fault in Japan has a recurrence interval of 14-15 ka (Kumamoto, 1998). Indeed, further studies could
reveal longer intervals, but even doubling or tripling this value limits the quiescence period within the Holocene and the Late
Pleistocene. Even in slow deforming continental regions, substantiated recurrence intervals fall within 20–30 ka (Bollinger et
al., 2021). This estimate is close to Trifonov and Machette (1993) estimate for the World Map of Major Active Faults. Late
Quaternary deformations remain on the Earth's surface and could be unambiguously distinguished when found at late
Quaternary landforms. Therefore, remote sensing interpretation of recent landforms is sufficient for regional-scale mapping or
paleoseismological fieldwork planning.

Identifying and mapping active faults must precede any detailed research, and many studies report results of sole remote sensing interpretation. We account for such kind of information after the verification following guidelines discussed by Trifonov and Kozhurin (2010): a studied object is unlikely to be created by sole non-tectonic surface processes; it offsets landforms in the same direction, which is consistent with the fault pattern of the region, and the offset is larger on older landforms.

In the database, authors make efforts to keep a balance between a unified representation of data and the intention to store all available data. Therefore, we propose a database model capable of indicating cases of a significantly different approach in studies.

## 3 Database Model

The Active Faults of Eurasia Database is a geodatabase in the shapefile format, an open standard for geospatial vector data. It
stores the spatial location of fault lines on the Earth's surface and their attributes (Fig. 2). Within the attribute table, field order streamlines the workflow of the database population, so the fields organically form two groups. The first group contains fault parameters transferred from sources "as is" and is accompanied by a complete bibliographic reference. Unification was applied to these data to ensure the uniform glossary, spelling of fault names, and spatial detail. The attributes of this group have a broad domain of values that may hamper querying but allow textual analysis. The second group of attributes, generated by the
database owners, provide consistent nomenclature to simplify GIS processing and querying.

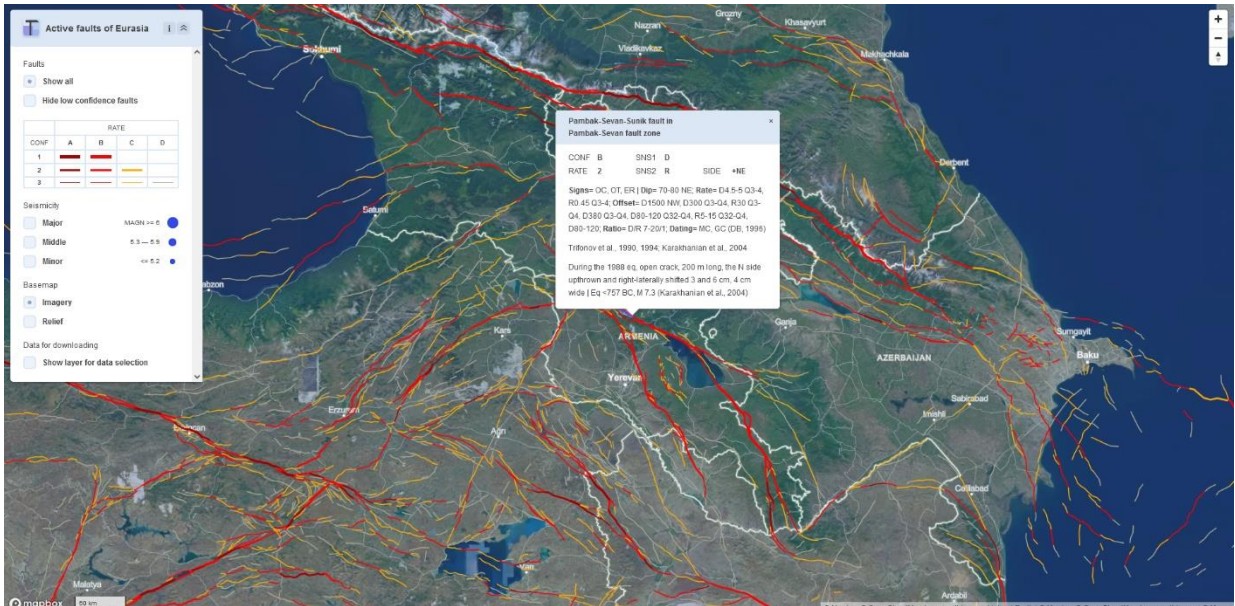

**Figure 2. Spatial pattern of AFEAD faults in Transcaucasia with an attribute table for an object within the Pambak-Sevan fault. Snapshot of the AFEAD web-interface at http://neotec.ginras.ru/index/mapbox/database_map.html (last access May 05, 2022).**

### 3.1 Geometry

Every AFEAD record has a two-dimensional linear shape stored as a polyline. In most cases, it represents a fault line crossing the Earth's surface and traced by scarp or a linear deformation of landforms, whereas some objects represent presumed intersection of a fault plane with the Earth's surface when an active fault was revealed by geophysical or seismological studies below a fold at the surface. The spatial data populated from sources is processed to comply with the topography, the database model, and target map of scale 1:1M. Usually, active fault deforms Earth's surface and may be traced on satellite imagery and

digital elevation models. We utilized the STRM global digital elevation model and Landsat 7 ETM+ imagery for georeferencing and spatial adjustment of published maps. In Soviet publications, the geographic location of studied objects was classified, so that processing of such data required deeper investigations and even reinterpretation of remote sensing data. To provide an accurate presentation at target scale of 1:1M, all the objects in the database were redrawn at a scale of 1:500 000 with uniform spacing between vertices whenever possible, disregarding geometry accuracy in source data. Most of the

fault lines bear attributes varying along it. In such cases, fault lines were split at the point where any attribute changes its value.

### 3.2 Attributes Transferred from Source Data

Collected data are stored in five fields: AUTH, FAULT_NAME, ZONE_NAME, PARM, and TEXT. Their order provides a workflow for data population, starting from identifying references and ending with auxiliary text data. This group of fields is supposed to store as much relevant data as possible and retain authors' interpretation. Still, some unification is applied to

ensure the consistency of definitions and naming. If studies provide contradictory data, all the provided parameters are recorded with a relevant citation. The low formalized string fields of this group are designed to store different estimates of the same parameter and relevant references; any normalized representation of these parameters would either eliminate the context or increase the complexity of the database dramatically. As a shapefile may store up to 254 symbols in a field, exceeding the field length is resolved by omitting the least relevant data. As the present-day active fault studies provide rare cases of abundant yet

different estimates of fault parameters, the length limit does not hamper the use of the shapefile standard.

    **AUTH** is a non-nullable field that stores brief English references to the studies considered that object. The field format complies with the reference list (see Sect. 3.4) for further bibliographic retrieval.

    **FAULT_NAME** and **ZONE_NAME** store proper names of a fault and a fault zone, if available. In most cases, a fault zone includes a group of faults, either named or nameless. Name of a zone may originate from the name of its main fault, and an

object with equal values of FAULT_NAME and ZONE_NAME should be interpreted as a main fault within a zone. If a fault has an ambiguous attribution to several fault zones, all of them are listed in ZONE_NAME, comma-separated. Uniform name spelling and designation of objects to a zone or a single fault are maintained in the database, even if varying among studies.

    **PARM** is a formal description of fault parameters. It has a dictionary-like structure (Fig. 2, Appendix) with mnemonic keys ending with "=" and a comma-separated list of values after it. A sequence of keys from a single source is separated by

semicolon, whereas different authorship is indicated with a vertical bar "|" and provided with the reference. For brevity, units

of measurement are omitted with length reported in meters, depth in km, rates in mm/yr, age in years BP or units of the geological time scale (e.g., N1 for the Miocene, Q4 for the Holocene), location in decimal degrees, unless otherwise stated explicitly. Acronyms are applied for directions and sense of slip (Table 1, Appendix). "Signs=", and "Dating=" have coded values, the complete form provided in Appendix. These keys originate from DB96 attributes (Trifonov and Machette, 1993), although they are concatenated into a single synthetic field. This denormalization was applied as separate attributes proved ineffective due to the excessive number of null values and an occurrence of valuable yet different estimates of the same parameter.

**TEXT** contains free-form comments on the fault characteristics supplementing other fields of this group.

**Table 1. Acronyms for fault sense used in the database.**

| | |
|---|---|
| **N** | **N**ormal fault |
| **R** | **R**everse fault |
| **T** | **T**hrust fault (reverse fault dipping <45º) |
| **D** | **D**extral (right-lateral) strike-slip fault |
| **S** | **S**inistral (left-lateral) strike-slip fault |
| **E** | **E**xtensional fracture |
| **V** | Sense unknown, **v**ertical offset |
| **U** | Slip **u**nknown, no vertical offset |

**3.3 Derivative Attributes**

The derivative attributes are those produced by the database owners based on collected data. Data domain is defined for the fields of this group (Table 2), thus providing a basis for classification and spatial analysis. All of them, except for SENS2 and SIDE, are non-nullable.

**Table 2. Values of derivative attributes.**

| Attribute | Domain | Comments |
|---|---|---|
| FAULT_ID | integer | Unique ID number of the object |
| RATE | {1, 2, 3} | 1 – high, estimated rate exceeds 5 mm/yr, 2 – medium, estimated rate 1 to 5 mm/yr, 3 – low, no estimates or estimated rate below 1 mm/yr |
| SENS1 | {N, R, T, D, S, E, V, U} | Described in table 1 |
| SENS2 | {N, R, T, D, S, E, V, null} | Described in table 1 |

| UPSIDE | {N, NE, E, SE, S, SW, W, NW, null} | Direction of the uplifted side |
| --- | --- | --- |
| CONF | {A, B, C, D} | A – proved active by a series of published either historical and instrumental evidence or paleoseismological studies; <br> B – unambiguous surface deformations, but no recent slips have been described yet; <br> C – few pieces of evidence of activity are known; <br> D – fault once stated active, but the evidence is insufficient or even absent. |

**CONF** indicates a level of confidence that a particular object once identified as an active fault meets the conventional definition of the active fault. Its variations represent the fact that the definition of the active fault varies between studies depending on their objectives and reasoning behind active fault mapping. In AFEAD, by "confidence" our team means a qualitative measure of expectance that an independent researcher would support the hypothesis of fault activity considering evidence from sources provided in the AUTH field. We are aware that it is a highly ambiguous criterion, still being a crucial attribute of an active fault, as this concept varies in meaning significantly among studies. Confidence in activity is unrelated to age or level of source publication and cannot be technically derived from it. Following guidelines have been proposed for CONF values assignment:

A – proved active by a series of published either historical and instrumental evidence or paleoseismological studies, multiple crustal earthquakes occurred at the fault line consistent with fault sense;

B – unambiguous surface deformations, but no recent slips have been described yet, attribution of earthquakes is questionable;

C – few pieces of evidence of activity are known, lack of seismicity, surface deformations, or both;

D –once declared active, but the evidence provided for such opinion is insufficient or even absent.

CONF is reconsidered if new data are obtained for the studied object as well as if new data for neighboring faults support or doubt the activity of the studied object.

**RATE** is a rank of deformation rate with estimated boundary values of one and five mm per year, which may not be considered as actual measured slip rates. Slip rate is unlikely to be justified well at low confidence faults so that RATE is meaningless for such faults. Therefore, boundary values of RATE have been chosen to affect primarily high-confidence faults with minimal rate estimation error. However, measurement techniques and their accuracy may vary greatly even in well-studied regions, thus making RATE a qualitative indicator acceptable for regional-scale visualization and spatial analysis. RATE source data are point measurements stored in the PARM field of the relevant fault segment; thus, RATE is propagated along the fault line considering topography. Even though this approach implies ambiguous rank assignment, it is the most effective way of representing rather scarce rate data.

**SENS1** and **SENS2** represent sense slip motion by its primary and secondary components (Table 1); SENS2 has values only for faults with identified oblique slip. Objects with poorly constrained slip are identified as SENS1 = "U" (for "unknown") or SENS1 = "V" (for "vertical") where vertical offset is evident along the scarp.

**UPSIDE** is a direction of an upthrown side of the fault recorded as cardinal direction.

**3.4 Reference List**

Most of the entries in the database have been collected from published studies, 621 references and 63 unpublished pieces of data. The reference list represents the database sources as a tab-separated text file with citations equal to the domain of the "AUTH" field and a complete bibliographic reference to the source publication in English and in the language of publication. The reference list is supplemented with a relational table many-to-many that connects FAULT_ID with primary keys of the

bibliographic reference table, a "citations" field. AFEAD references to the initial studies, even though they may be considered outdated, and to the studies that provided significant contributions to each object. For data compiled for the World Map of Major Active Faults (Trifonov and Machette, 1993; Trifonov, 2004) and never published before it, the reference is supplemented with the name of a researcher responsible for a region containing the fault (e.g., Kozhurin, data 1996). Recent unpublished contributions have reference to the responsible researcher (or "working material" for contributions of the database

owners) and the year of update.

**4 Source Data**

The actual structure and contents of AFEAD is defined by a style of data presentation in recent active fault studies. AFEAD generally inherits the approach of the World Map of Major Active Faults (scale 1:5M, Trifonov and Machette, 1993), although being radically improved in spatial detail and fault parameterization. Those improvements required the compilation of

heterogeneous sources, mostly regional maps of active faults and research papers. Due to the amount of processed data, we update any fault zone just to the target detail and resume the processing in rare cases of contradiction between the database and newly published data. We exploited several sources of fault characteristics with methods of data processing vary between them and present them following the workflow of data population in AFEAD.

The first attempt to collect a global-scale inventory of active faults was the World Map of Major Active Faults that was initiated

in 1989 and approved by the International Lithosphere Commission in 1990 as the Project II-2 of the International Lithosphere Program (ILP). The Project was supported by UNESCO as a contribution of the ILP to the UN International Decade of Natural Disaster Reduction and was included in the ILP Global Seismic Hazard Assessment Program in 1993. During that project, the AFEAD authors proposed data representation methods (Trifonov and Machette, 1993) and the model of database DB96 for parametrization of active faults (Ioffe and Kozhurin, 1996) based on the software design of Ioffe et al. (1993). This first

database of active faults of Eurasia contained about 10,000 objects. These data were published as the active fault maps of Eurasia and Northern and Eastern Africa, 1:10,000,000 (Fig. 3, Trifonov, 2004), Eurasia, 1:5,000,000 (Trifonov, 1997), and in the table format (Trifonov et al., 2002).

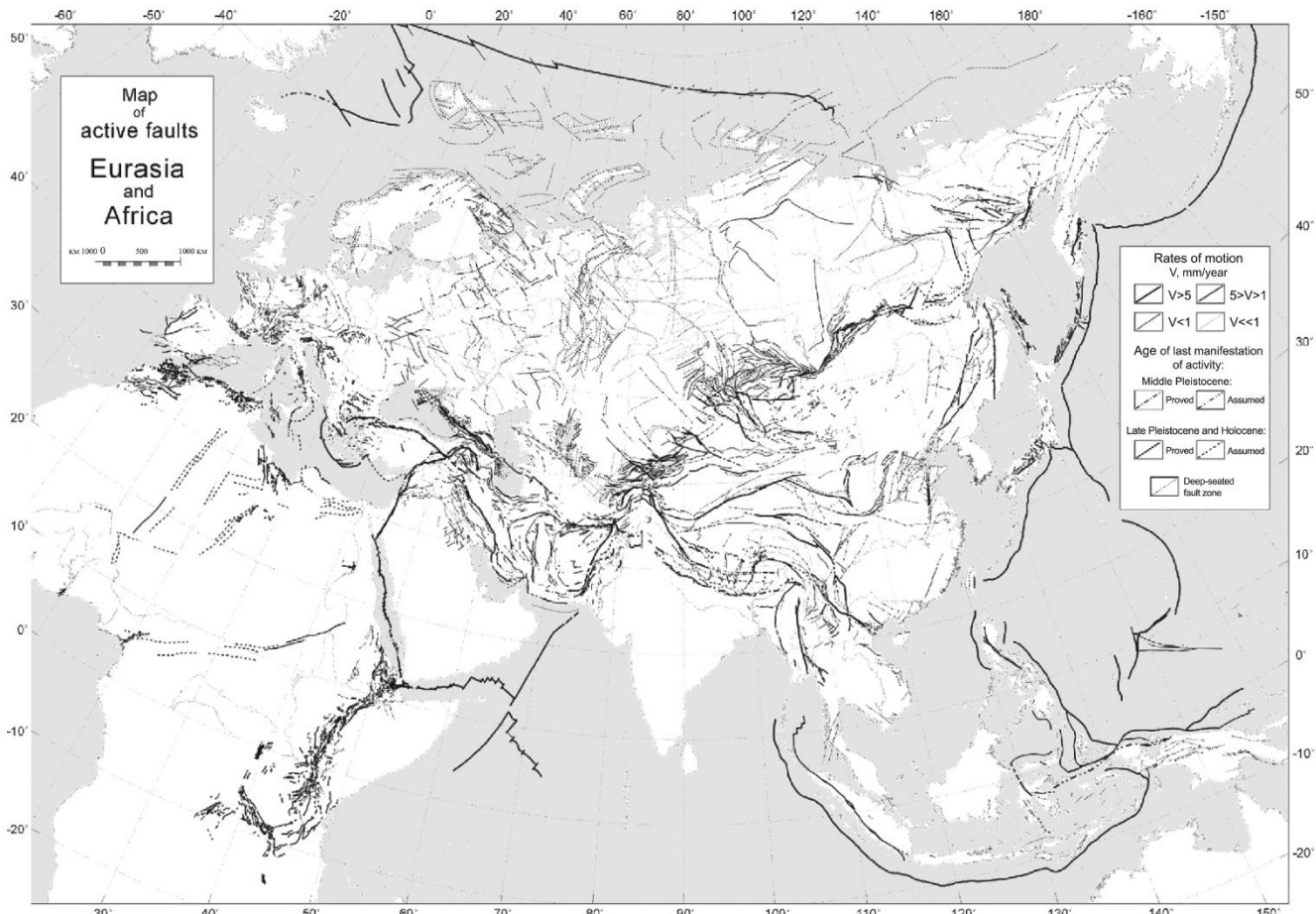

**Figure 3. A selection from the World Map of Major Active Faults for Eurasia and Africa (Trifonov, 2004).**

In AFEAD, we use the spatial database of the World Map of Major Active Faults as a base layer, referred to as DB96. Its entries were redrawn to comply with the target scale, the attributes were updated according to the actual database scheme and the references were revised, thus providing a framework for further data collection. During AFEAD development, much of DB96 data have been replaced by more relevant information. However, there are some regions that cannot be updated due to the absence of studies after the publishing of DB96.

Among the rest of the sources, a large amount of data was collected from region-scale maps and databases of active faults (e.g., Hessami et al., 2003, Basili et al., 2013), both digital and published in hardcopy. Such maps usually bear sufficient information on slip direction and rate as well as generalization level but lack reasoning. Therefore, their processing includes georeferencing, verification of their spatial location against topography, and population of attributes (Fig. 4). CONF is to be set uniform and low, and its elevation requires additional studies at individual faults.

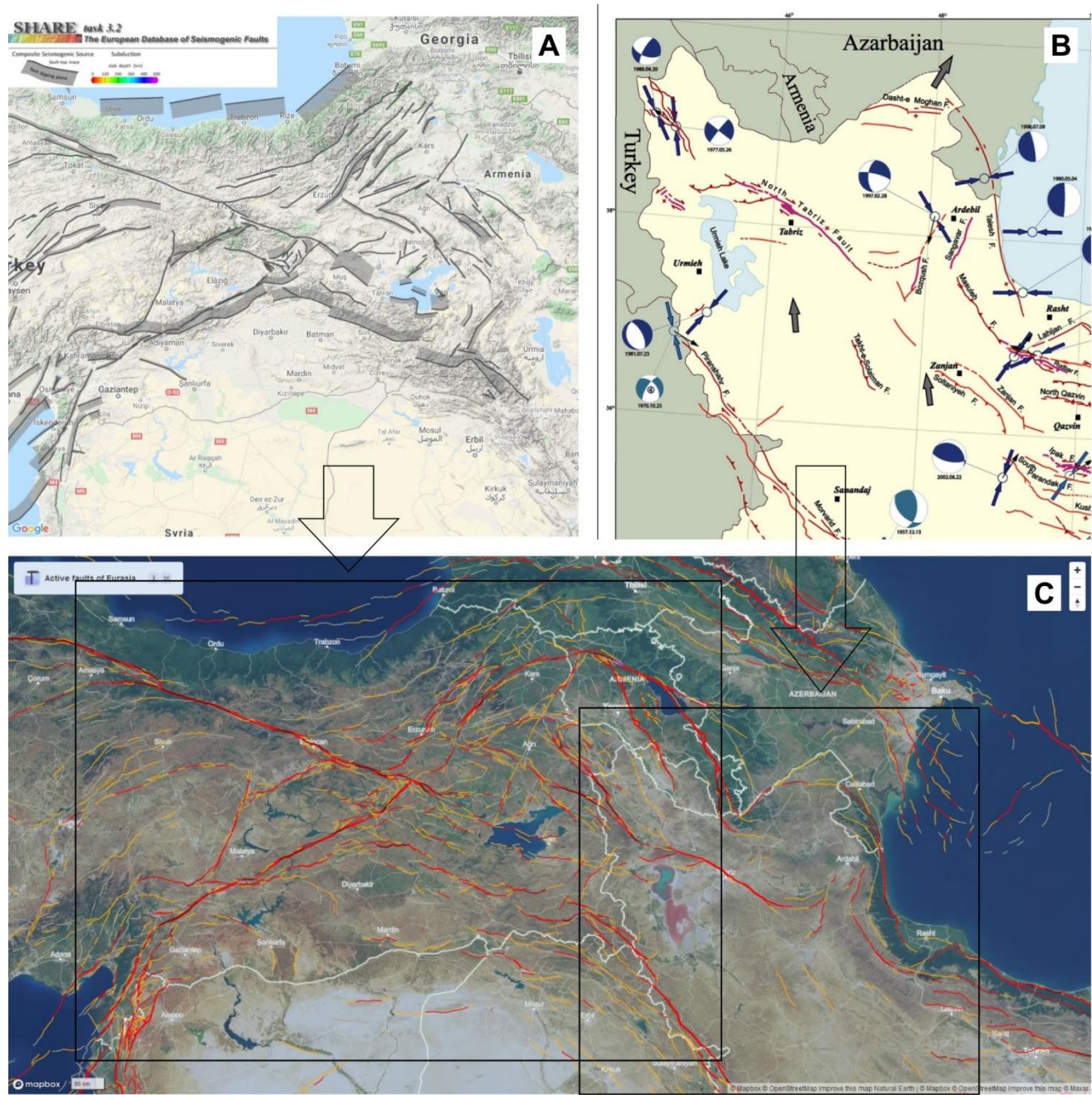

**Figure 4. Active faults in Transcaucasia: A, web-interface of the European database of Seismogenic Faults (Basili et al., 2013); B, a fragment of the Major Active Faults of Iran Map (Hessami et al., 2003); C, AFEAD web-interface (http://neotec.ginras.ru/index/mapbox/database_map.html, last access May 05, 2022) for this area.**

Research papers are the most comprehensive sources considering a particular fault or a fault zone. A standard structure of a research paper provides both parameters of active faults and rationale for it, thus allowing us to assess the reasoning and choose

the most relevant and accurate data among published. However, the methods and definitions significantly vary in sources. Therefore, they require a thorough analysis and normalization before populating (Fig. 5).

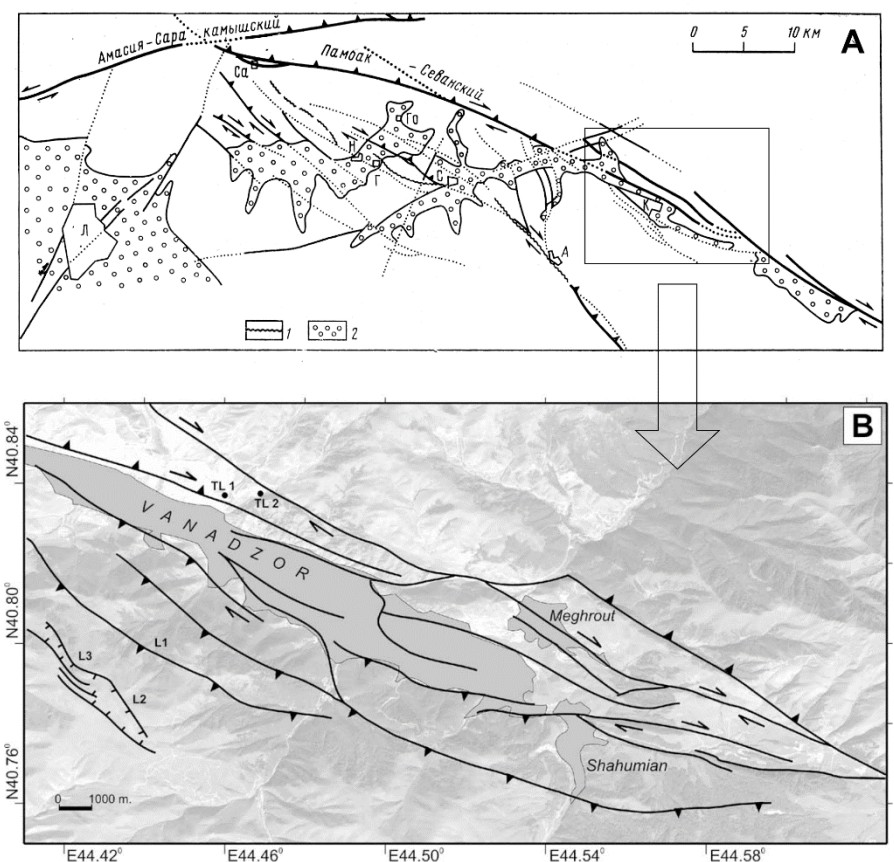

**Figure 5. Sources for mapping Pambak-Sevan fault near Vanadzor city, Armenia. A, an initial study of Trifonov et al. (1990), geographical objects and coordinates had been omitted due to Soviet legal requirements; the location of Vanadzor is labeled with K (for its former name Kirovakan) within the box. B, a more recent georeferenced map (Karakhanian et al., 2004). Both papers affected the location and attributes of the Pambak-Sevan fault in AFEAD, including the object highlighted in Fig 2.**

Spatial adjustment and verification of populated fault parameters require complementary sources. The global digital elevation model SRTM V3 and Landsat 7 ETM+ imagery serve as base maps for any spatial processing. Earthquakes are strong evidence of fault activity, so we utilized catalogs of the National Earthquake Information Center (NEIC), U.S. Geological Survey (https://earthquake.usgs.gov/, last access May 05, 2022) and the International Seismological Centre (ISC, http://www.isc.ac.uk/, last access May 05, 2022) to refine the confidence in fault activity (a value of "CONF" field) and, where needed, to specify fault plane geometry and slip sense.

After the initial population of the database, new data may contradict with AFEAD. There is no direct relation between the recency of the information and its accuracy, so that any join of recent data requires comparison of reasoning behind older and recent objects. The result of the comparison affects CONF in either its elevation or decrease up to deletion from the database.

Published data have to be supplemented by additional interpretation of remote sensing data to facilitate the uniform detail of the database by adding unpublished objects. In most cases, additional research was required at the spatial boundaries of cited studies (e.g., national borders, limits of a tectonic structure). Unfortunately, a significant amount of fault zones still lacks published data. The absence of relevant published information could be confused with the absence of active faults, so the AFEAD authors collect unpublished data, after thorough consideration and assigning a level of confidence. This contribution significantly improves the spatial pattern of active faults in remote areas, mainly in North Asia. Such unpublished entries will be updated as soon as new studies are published, although it is unlikely to occur in the near future for all the entries.

## 5 Overview of the Dataset

### 5.1 General characteristics

The database comprises 48,205 objects – active faults and their segments. Most of them (ca. 44,000) belong to mainland Eurasia, whereas the rest are located on islands or underwater. Active faults tend to group in broad belts (Fig. 1) incorporating minor plates and crustal blocks at margins of tectonic plates. However, active faults were mapped elsewhere across tectonic plates as well. Individual faults are indistinguishable on the continent-scale map, but a fault density map (Fig. 6) shows an even greater contrast between active belts and cratons.

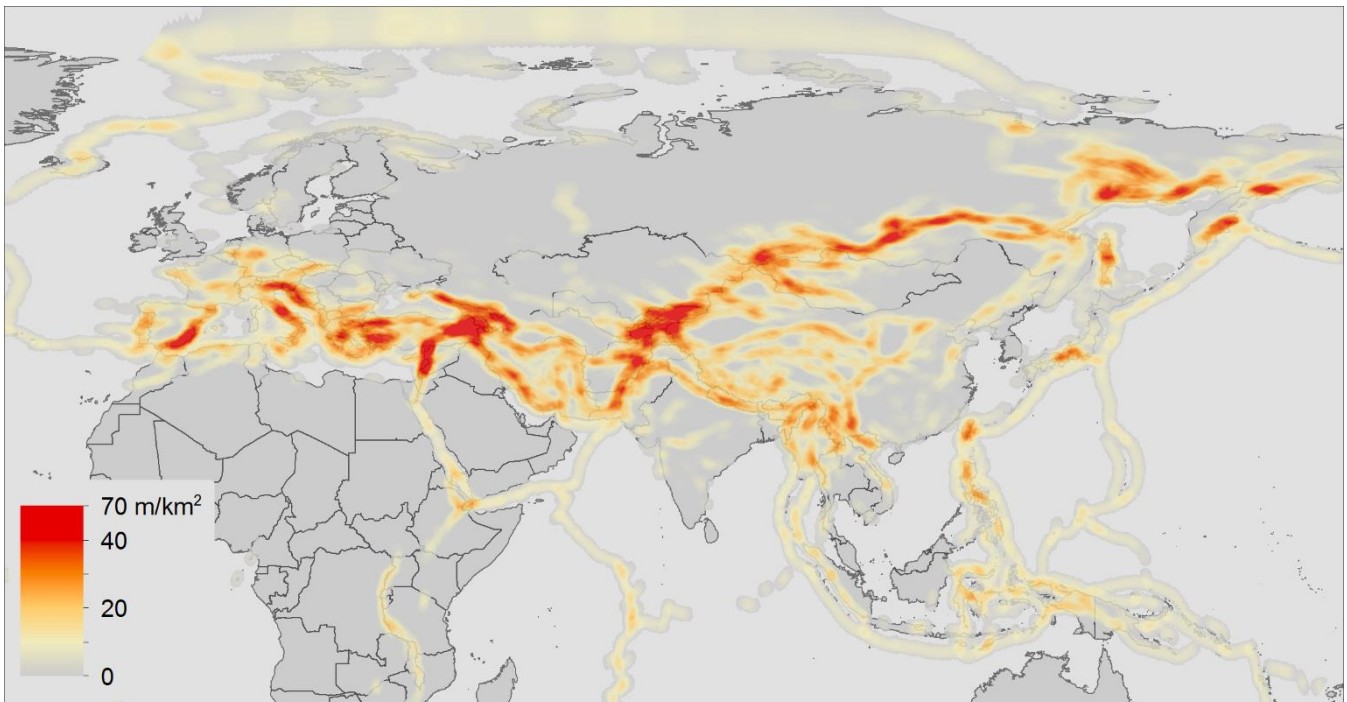

Figure 6. The density of active faults in AFEAD, meters per km². Low-confidence faults (CONF="D") are omitted.

The mean length of database objects is 22 km, and 90 % of them belong to the range 5-60 km (Fig. 7). Most of the faults longer than 100 km are underwater, where no detailed data are available.

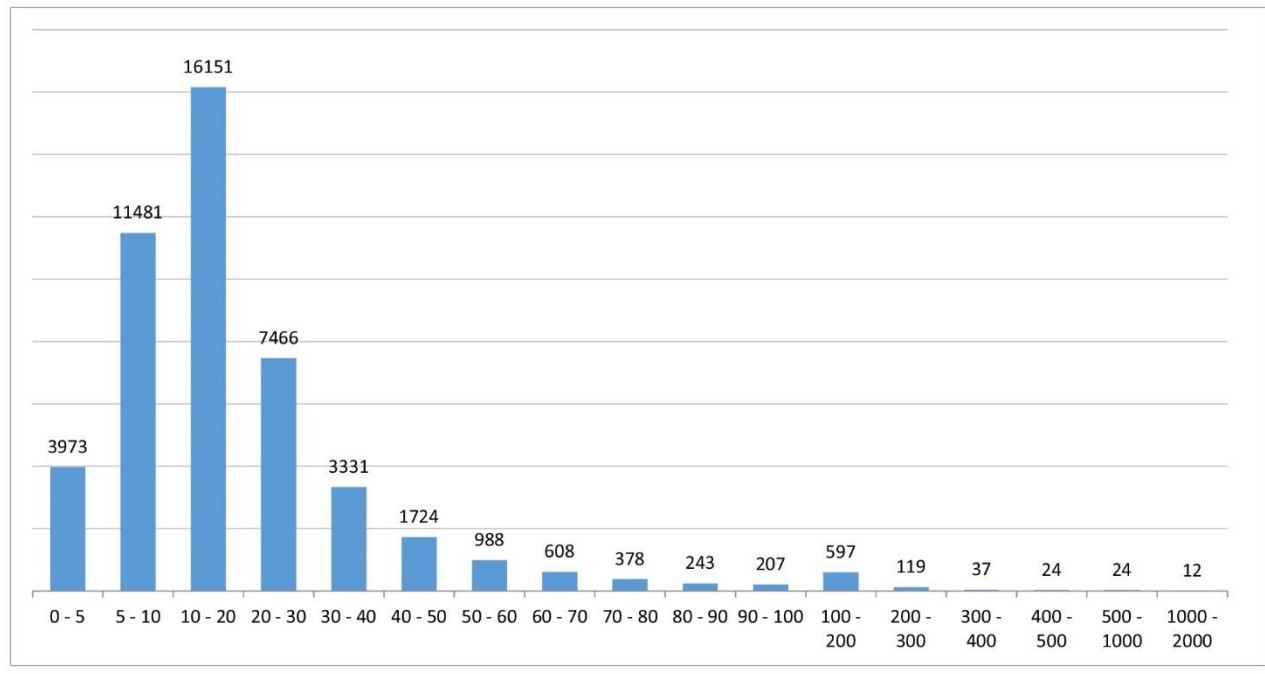


**Figure 7. Distribution of AFEAD objects by length.**

Within the dataset, 81% of objects have been published before, including 36% with two and more sources. The PARM field represents the amount of data yielded by studies. Aside from unpublished, 31% of entries have empty PARM, most of them originated from maps with no parametrization. Complete descriptions are common in the Western Pacific, Baikal Rift, and

Alpine-Himalayan Belt. in the latter, prominent clusters are located in Eastern Mediterranean and Anatolia, Transcaucasia, Iranian mountains, and Tibet region. On the contrary, active faults within cratons typically are poorly studied and low-confidence, thus representing different approaches of active fault studies.

The intensity of motions at a fault affects two parameters: RATE and CONF (Fig. 8). The former is a natural characteristic, whereas the latter represents the quality and quantity of studies considered the fault (Table 2). Therefore, subsequent researches

may elevate CONF, even at slow-moving faults. Most of AFEAD entries (94%) are slow-moving faults (RATE=3); all low-confidence (CONF="D") and most CONF="C" faults fall within this class. Deformation rates exceeding 1 mm/yr are identified for 6.2% of entries; faults exceeding 5 mm/yr comprise 1% of entries. The amount of both CONF="A" and CONF="B" faults in RATE classes gradually decreases from slow-moving to fast-moving.

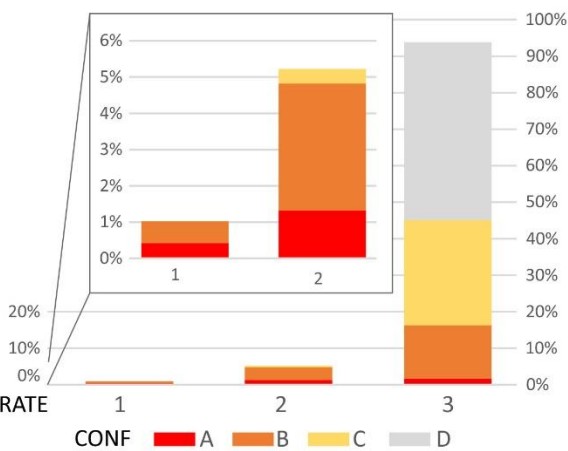

Figure 8. Distribution of AFEAD objects by classes of RATE (horizontal axis) and CONF (color). Inset, zoomed RATE classes 1 and 2.

The most frequent fault sense in AFEAD is reverse (21%) even considered separately from thrusts (7%), the normal sense is identified in 17% of objects, right-lateral and left-lateral faults are equally common (13% and 12%). Dip-slip is identified in 18% of entries, whereas the sense of slip remains unknown for the rest 11% of entries. A secondary component is provided for 22% of entries (Fig. 9).

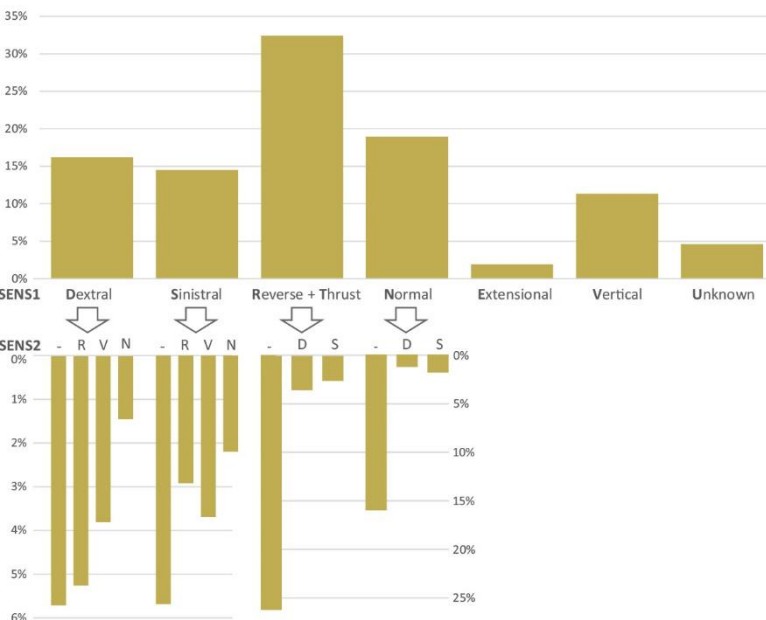

Figure 9. Distribution of AFEAD objects by fault sense: top, SENS1, bottom, SENS2 for each SENS1 value. For sense indices see Table 1.

## 5.2 Regional examples

To provide a deeper insight into the AFEAD structure and usability, we present two contrast examples from the compressional setting of the Caucasus and the transtensional setting of the Baikal region corresponding to 4°×6° map tiles K-38 Tbilisi (http://neotec.ginras.ru/index/datamap/AFEAD_K38_Map.html, last access May 05, 2022) and N-49 Chita (http://neotec.ginras.ru/index/datamap/AFEAD_N49_Map.html, last access May 05, 2022).

The Caucasus is located at the northern flank of the Alpine-Himalayan collision belt. It has been experiencing compression

since the Oligocene (e.g., Nikishin et al., 1998) that, together with mantle geodynamics, built up a high mountain range of the Greater Caucasus and highlands of Transcaucasia south of it. Recent deformation in the area is concentrated at the Main Thrust of the Greater Caucasus and an arcuate system of Zheltorechensky-Sarykamysh, Pambak-Sevan, Garni, and Hanarasar strike-slip fault zones in Transcaucasia (Fig. 10). Minor faults generally follow their pattern, although they are scattered across large areas up to the northern foothills of the Caucasus. In the Greater Caucasus, most of the faults had been identified by the 1980s

and published in monographs on broad geological topics (e.g., Milanovsky, 1968, Kogoshvili, 1970) incorporated into DB96. Few works were carried out after the compilation of DB96; thus, the Greater Caucasus area appears to lack state-of-art paleoseismological studies. Transcaucasia had been much less studied until the infamous 1988 Spitak earthquake (Ms =6.7, Bommer and Ambraseys, 1989). The subsequent extensive studies (e.g., Trifonov et al., 1990; Karakhanian et al., 2004) revealed spatial patterns and geodynamic settings of active faulting. Hence, AFEAD entries in Transcaucasia bear much more

attributes than those at the Greater Caucasus.

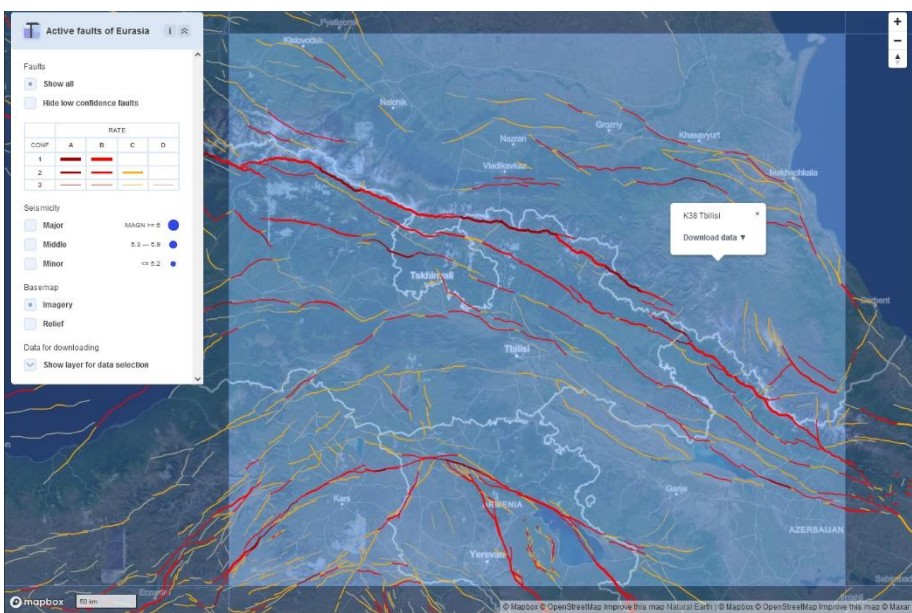

**Figure 10. Representation of active faults in the Caucasus by AFEAD web-interface (http://neotec.ginras.ru/index/mapbox/database_map.html, last access May 05, 2022).**

Another case of different tectonic settings and research history is the Baikal Rift zone (Fig. 11). It is a linear system of grabens bounded by normal or transtensional faults developing since the Oligocene (e.g., Logatchev and Zorin, 1992). General features of this zone had been identified by the 1980s (Sherman and Levi, 1978; Solonenko, 1977 Logachev, 1984), but intermittent seismicity keeps drawing constant attention to the active faults of the region. In addition to research papers, recent studies have been published as regional data collections: an inventory of paleoseismic sites by Smekalin et al. (2010), the database of Pliocene-Quaternary faults of Southern East Siberia (Lunina et al., 2014), and the Map of Seismotectonics of Eastern Siberia (Imaeva et al., 2015). A compilation of these sources provides uniform input for the AFEAD.

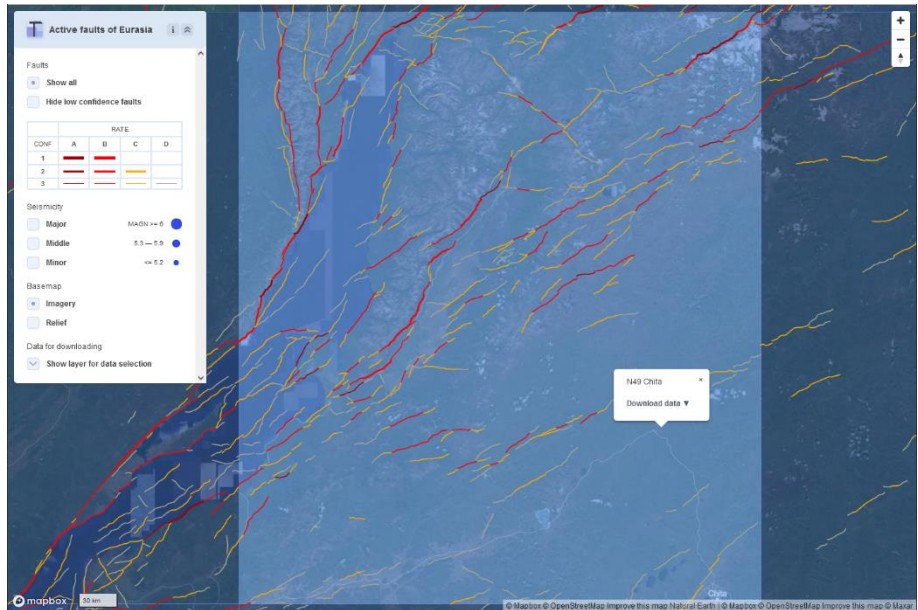

**Figure 11. Representation of active faults at the Baikal rift zone by AFEAD web-interface (http://neotec.ginras.ru/index/mapbox/database_map.html, last access May 05, 2022).**

**6 Update Strategy**

Since the initial populating of the database, the database owners have been monitoring scholarly literature for relevant data. After the acquisition of new data, its reference is appended to the reference list and a graphical representation of faults is georeferenced and traced, if needed. Their location is checked against AFEAD fault patterns, so that the location of the new fault line may either differ from AFEAD objects or match some of them. In the first case, the fault line is redrawn to comply with the topography and target map scale of 1:1M, and primary attributes are populated: AUTH, FAULT_NAME, ZONE_NAME, PARM, and TEXT. Derivative attributes of kinematics (SENS1, SENS1 and UPSIDE) are set according to them and topography. Confidence of activity relies both on reported evidence of activity (Table 2) and on coherence of fault parameters with parameters of adjacent AFEAD objects. If data for neighboring faults support the activity of the studied object, it may elevate CONF for the entire zone. Otherwise, if faults contradict each other, CONF is reduced for the new fault or pre-existing objects, or both, depending on the reasoning behind each object. Finally, RATE is set according to the actual slip

measurement at the fault and its confidence, if possible, or propagated from other objects in the fault zone. The second case of matching locations causes an update to the existing AFEAD object. Its location is then adjusted according to the reasoning in the new source and AFEAD. The new citation is then appended to AUTH, and newly acquired parameters are appended to PARM as well as comments to TEXT. If these affect attributes of kinematics SENS1, SENS1 and UPSIDE, their values are reconsidered. New reported evidence of activity may elevate CONF value as well.

The presented database AFEAD v.2022 has reached target detail, and no major revisions in the database model are planned after the completion of this study. However, new versions will be released after the acquisition of recently published data. To ensure data consistency, no direct external contribution to the database is possible. The authors encourage researchers to inform us about missing or recently obtained data via the e-mail of the corresponding author.

## 7 Data Access

The main access point to the most recent version of AFEAD is a web map available at http://neotec.ginras.ru/index/mapbox/database_map.html (last access May 05, 2022). The current data set v.2022 is available at https://doi.org/10.13140/RG.2.2.25509.58084 (Bachmanov et al., 2022). A variety of up-to-date database representations, supplemented by the reference list and explanatory notes, are hosted at http://neotec.ginras.ru/index/english/database_eng.html (last access May 05, 2022); it includes a raster overview map, raster map tiles designed for print, .kmz, and .shp vector tiles.

Studies considering the AFEAD scheme or its reasoning may refer to this study, DOI of the database, or the initial publication on this topic of Bachmanov et al. (2017). None of them would be an acceptable reference for studies considering fault locations or their parameters; instead, the researcher is advised to cite source studies provided in the AUTH field.

## 8 Conclusion

  AFEAD is the largest and the most comprehensive collection of active faults comprising ~48,000 entries spanning entire
Eurasia and adjacent seas. For each entry in the database, its spatial location and characteristics of motions are provided. All spatial data have uniform detail equal to hardcopy maps of scale 1:1M. Attributes of faults store relevant information transferred from sources and derivative parameters generated by the database owners.

  The database makes possible a spatial search for local studies. It provides sufficient detail for planning a study of a particular fault system and guides deeper bibliographical investigations, if needed. This scenario is particularly significant for vast
Central and North Asia areas, where most studies are available only in Russian and hardcopy. Moreover, the database model provides the basis for GIS-based regional and continental-scale integrative studies. The authors suggest that the use of this database will support geodynamic and paleoseismological studies in Eurasia.

## Author contributions

EZ and DB conceived and set up the paper. DB, VT, and AK designed the database and compiled the data. SG contributed to database normalization and provided an overview of the dataset. All the authors discussed the study and contributed to the writing of the manuscript.

## Competing interests

The authors declare that they have no conflict of interest.

## Acknowledgements

The authors thank all the colleagues supported the database population. The contributions of N.N. Govorova and G.I. Volchkova made it possible to sustain and develop the database. We are deeply grateful to Dr. A.I. Ioffe, Dr. G.A. Vostrikov, and R.V. Trifonov for their contributions to the ILP II-2 database design strongly affected this study.

## Financial support

This research was supported by the Russian Science Foundation, grant no. 17-17-01073-p. State scientific programs have supported data gathering since 1990s, ongoing project no. 0135-2019-0051.

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

**Appendix. Data domain of the PARM field**

**Age=** Age of the latest dated slip in years BP or units of the geological time scale (e.g., N1 for the Miocene, Q4 for the Holocene).

**Signs**= Signs of recent fault motion:

DF      Drape fold,

DT      Sharp change of recent deposits thickness,

EC      En echelon array of compressional structures,

EQ      Earthquake hypocenter,

ER      Surface seismic ruptures,

ET      En echelon array of extensional structures,

FD      Surface folding,

FM      Earthquake focal mechanism,

FR      Linear group of fractures,

GA      Gas and hydrochemical anomalies,

GD      Geodetic surveys,

GP       Geophysical data,

         HS       Historical or archaeological data,

         HT       Hydrothermal springs,

         LS        Linear group of landslides or rockfalls,

         MV      Mud volcanism,

OC       Offset river channels,

         OD      Offset recent deposits,

         OT       Offset river terraces or alluvial fans,

         PS        Paleoseismic sites,

         TS        Sharp change of tectonic structure

VC       Volcanic chain.

         **Dating**= Dating techniques:

         AR       Archaeological,

         CR       Radiocarbon,

         GC      Geological correlation,

HI        Historical,

         IN        Instrumental,

         LH       Lichenometry,

         MC      Geomorphological correlation.

         **Layers**= Faulted layers of the lithosphere:

S         Sedimentary cover,

         UC      Upper crust,

         LC      Lower crust,

         M        Upper mantle.

         **Dip**= Dip angle (in degrees), dip direction (cardinal directions), occasionally supplemented with site coordinates or fault part
(cardinal directions, C for central) for which data are relevant.

         **Depth**= Fault depth in km.

         **Offset**= Measured offset at the surface, occasionally supplemented with site coordinates or fault part (cardinal directions, C
         for central) for which data are relevant.

         **Rake**= Rake, the angle between the slip direction and the strike line.

**Rate**= Average slip rate, mm/yr, supplemented with time span and occasionally site coordinates or fault part (cardinal
         directions, C for central) for which data are relevant. An asterisk (*) indicate geodetic measurements, a double asterisk (**) -
         seismological measurements.

         **Ratio**= Ratio of strike slip and dip slip.

**Seism**= Parameters of an earthquake occurred at the fault: magnitude, (name,) date, depth.

**SeismDepth**= Depth range of earthquakes at the fault.

**SeismRecur**= Mean recurrence interval of earthquakes at the fault.

**Sense**= Fault slip sense, abbreviated (as in Table 1).

**Side**= A direction of an upthrown side of the fault.