# Peer review of "The Active Faults of Eurasia Database (AFEAD): Ontology and Design behind the Continental-Scale Dataset"

_Earth System Science Data, 2021_

## Author Response (AR1)

We have revised the manuscript based on the suggestions and advice of the reviewers. The authors are very grateful for the reviews, as they have improved our manuscript considerably. An item-by-item response to their comments is enclosed. We hope that these revisions successfully address their concerns and requirements. We have enumerated all the referees' comments in the following answers to streamline the review process.

**Dr. Alessandro Tibaldi:**

*1.1. I would only encourage the authors to add some more explanations about the seismicity showed in their web map.*

- An explanation has been added at the "seismicity" block of the web map interface according to the preceding author comment AC1: "Crustal earthquakes that occurred at the AFEAD faults and recorded with the key "Seism=" in PARM field. Sources: National Earthquake Information Center (NEIC), U.S. Geological Survey and the International Seismological Centre (ISC)"

*1.2. I have seen the lacking of some major active faults in the present database; for example, in Iceland very few faults have been presented in the database, respect to the available information.*

Fault pattern in Iceland was considered detailed enough during the database population. However, provided data is included in the forthcoming update of the AFEAD v.2022.

*1.3.*

The other comments suggest straightforward corrections to the text, and we accept them.

**Anonymous Referee #2**

*2.1. Scientific weaknesses*

*2.1.1 The data collection is based on bibliographical investigations, but most of the bibliographic references are quite outdated. Out of the 657 references (in the Excel file), only 13 are post-2010. Of these 13, three are classified as unpublished information. Of all 657, 55 are classified as unpublished information, most of which are as old as 1996. How reliable can be a piece of information supplied to the authors 25 years ago and never published since then?*

Indeed, old and unpublished information is the least reliable source. Unfortunately, those cases cannot be considered outdated *sensu stricto* due to the absence of more relevant information. We are grateful that the referee highlighted this topic, but cannot agree that it is a scientific weakness of the database; instead, it displays a bias in active fault studies towards most active or easily accessible fault systems. Referee's concerns on the reliability have already been accounted for in the CONF (level of confidence) parameter.

*2.1.2. In the last decade, several active fault databases have been published containing updated information. Below I list some of them (not necessarily exhaustively) that have significant geographical overlap with AFEAD and contain more up-to-date data than AFEAD:*

Provided data will be included in the forthcoming update of the AFEAD v.2022; a portion of data has been already populated after the AFEAD v.2021 release. However, they are not comparable to AFEAD by extent or detail or both.

*2.1.3. Apart from those compilations released in the last year, most of these have been around for quite a long time now. In addition to this lack of data, the relationship between the fault representation in*

*AFEAD and the fault representation in the source dataset is not clear. This is of particular concern for the blind faults since only criteria associated with the topographic signature are recalled. On the one hand, not considering the latest fault compilations prevents AFEAD from listing the newly recognized active faults. On the other hand, it also prevents AFEAD from eliminating those faults that were once considered active but are currently considered not active based on new evidence. Unfortunately, the CONF parameter does not consider the recency of the information.*

A workflow of transferring source data to the AFEAD representation is presented in section 4. Source Data. We have expanded this section to clarify the workflow, especially in the cases of contradiction among data sources. There is no direct relation between the recency of the information and its accuracy, so any join of recent data requires a comparison of the reasoning behind older and recent objects. The result of the comparison affects CONF in either its elevation or decrease and even deletion from the database.

*2.1.4. The compilation of the fault parameters also remains rather obscure in several aspects. For example, of the 47,363 faults, 22,270 (47%) have no parameter assigned (field "Parm" is NULL). Of the 25,093 faults with the field "Parm" not NULL, only 6,849 reports a "Rate=" value; how was then the Rate (rank) parameter assigned to the remaining faults?*

Objects of null "Parm" are typically those collected from fault maps with no parameterization. Please note that RATE=3 means "no measured rate above 1 mm/yr" (see Table 2), so it addresses all those cases.

**2.2. Technical weaknesses**

*2.2.1. The AFEAD is distributed as a single shapefile. Technically speaking, it is not even a database apart from the implicit relation between geographic features and their attributes. No relational table is provided between AFEAD and any of its linked information. In other words, it should be classified as a geographical flat-file, not a proper database.*

According to Wikipedia, "A database is an organized collection of data, generally stored and accessed electronically from a computer system." (https://en.wikipedia.org/wiki/Database), and AFEAD satisfies this definition of a database. However, it may not meet the definition of a relation database. Depending on the editor's decision, we can identify AFEAD as a "dataset" as it affects neither its inner structure nor representation. However, our experience in hosting and distribution of tectonic data shows that user-friendly shapefile format gets better reception among the researchers. Most AFEAD use cases require basic spatial analysis and text search on the user device without DBMS software.

*2.2.2. The fields in the shapefile attribute table are very poorly organized. First of all, none of the fields can be identified as a primary key. The lack of a primary key prevents the user from uniquely identifying any records and establishing their possible relations with external information. Also, the user cannot make an explicit reference to an individual AFEAD record when using it, including this review.*

A primary key has been added (field "FID").

*2.2.3. Both the "Auth" and "Parm" fields contain long text strings that, in the next update, could become even longer and easily exceed the limitations imposed by the shapefile format. Notice that the maximum number of characters in a text field of a shapefile is 254, see Attribute limitations in ESRI documentation at: https://desktop.arcgis.com/en/arcmap/latest/manage-data/shapefiles/geoprocessing-considerations-for-shapefile-output.htm#GUID-A10ADA3B-0988-4AB1-9EBA-AD704F77B4A2 or https://support.esri.com/en/technical-article/000012081*

Even accounting for shapefile standard limitations, we consider it the best format to distribute among researchers in the field of active faulting. It requires no proprietary software but supports spatial analysis and data queries. Only few objects are close to the maximum string length in AUTH or PARM

and this could easily be resolved by removal of outdated or least relevant sources. In the current AFEAD schema, field limitations do not affect data presentation and usability.

*2.2.4. These two fields are also very difficult to explore, especially the Parm field that contains very heterogeneous parameters. This poor organization makes it hard for the user to use the database. For example, selecting the faults that have a certain "depth" information would require a very complex query, which would discourage the non-experts in SQL and expose the users to uncertain results. Also, the Parm field takes up more bytes than needed by repeating within the field the word to identify the parameter type, such as "Sense=" or "Rate=" or "Depth=", occasionally also including the reference to the parameter itself.*

Indeed, PARM is designated for ease of reading, not querying. Below, the reviewer proposes to "separate the "Parm" attributes into different columns, paying attention to storing single numerical values in individual columns." A schema of the spatial database of the World Map of Major Active Faults (DB96) was exactly what the reviewer suggest, and we intentionally changed this approach in AFEAD. The suggested schema leaves no room for different estimates of the same parameter and references for these estimations. A defined domain of values will distort citing of data (e.g. single numerical value is required where only value range or upper estimate is known). Finally, well above 90% of such fields will be empty, which hampers visual interaction with data. However, if any parameter, e.g. depth, becomes credible for a large amount of data, it will be recorded to an individual column (say, DEPTH), like it was done for fault sense (fields SENS1, SENS2) and uplifted side (field SIDE).

*2.2.5. The use of the "+" (plus) sign in the "Side" field is unnecessary because all the non-null values are a plus. It could also be troublesome because the plus sign can be automatically converted when importing the data in other systems (try saving the attribute table into the Microsoft Excel format, for example).*

SIDE is a text field, and any DBMS may handle mathematical symbols in text strings.  We were unable to reproduce problems when opening .dbf attribute table in MS Excel. In active faults databases, it is common to label a downthrown side as well, so the plus sign serves as a reminder about an uplifted side.

**2.3. Other issues (listed by line "L" number)**

They suggest straightforward corrections to the manuscript, all of them were accepted. In AFEAD, strike-slip with unknown sense is considered equal to unknown sense (SENS1=U).

**2.4. Recommendations**

*2.4.1. The following technical fixes are necessary to make AFEAD suitable for using it in a proper DBMS.*

We consider shapefile to be the most suitable data format for the distribution of AFEAD at the moment. The provided guidelines will be essential for a redesign of AFEAD when demand for relation database managed by DBMS software increases.

*2.4.2. The European plate boundary along the Mid-Atlantic Ridge should be completed to make AFEAD adhere to its name (it could be disappointing for the AFEAD user to find data in the African plate and not the complete European plate).*

Faults in the Mid-Atlantic Ridge will be included in the forthcoming update of the AFEAD v.2022

*2.4.3. More explanations are needed to make the user understand the source of information used to assign the Rate ranks.*

Explanations have been added to the manuscript and AFEAD web map interface.

*2.4.4. A justification is needed for not considering all the recent fault data compilations published in the last decade. The authors should also discuss the implications due to the lack of updated information and warn the users about the limitations in using AFEAD instead of more up-to-date regional/local data.*

Explanations have been added to the manuscript and AFEAD web map interface.

---

## Referee Report (RR1)

Review of the manuscript "The Database of the Active Faults of Eurasia (AFEAD): Ontology and Design behind the Continental-Scale Dataset" submitted to Earth System Science Data by Egor Zelenin, Dmitry Bachmanov, Sofya Garipova, Vladimir Trifonov, and Andrey Kozhurin.

Black is the text of the first review.

*Italic-Blue is the text of the author's replies.*

Red is the text of this second review.

This manuscript describes the schema and strategy for compiling the Active Faults of Eurasia Database (AFEAD). It also provides a link to the database itself. The database can be accessed freely through a wed mapper interface and downloaded as images (jpg) with topographic background or as vectors (kmz or shapefile) of predefined tiles. The shapefile of the entire collection of faults and an Excel file with the list of references are also available for download through the ResearchGate link.

I commend the authors for the great effort in putting together such an extensive compilation of faults. I am also aware that there is a need for earth scientists to get hold of this type of data through a single access point. Nonetheless, I'm afraid that at the moment, this collection of data suffers from a few weaknesses. In brief, they are: 1) most scientific content is outdated; 2) the database design and organization of the data is technically poor. I elaborate on these aspects in the following.

This is my second review of this manuscript. I acknowledge that the authors tried to answer all the posed questions. Quite disappointingly, however, very few of the raised issues have been properly addressed. In most cases, the answers were dismissive, and the authors corrected none of the major issues. Nor did the authors provide a convincing justification in their replies. In a few cases, even when acknowledging the pointed-out weakness, the authors seemed to feel replying to this reviewer more urgent than strengthening the information to convey to their potential readers and database users. In summary, the authors elevated a major revision to a minor revision. Therefore, my conclusion remains the same as the first review.

More details on specific aspects are given below.

**Scientific weaknesses**

The data collection is based on bibliographical investigations, but most of the bibliographic references are quite outdated. Out of the 657 references (in the Excel file), only 13 are post-2010. Of these 13, three are classified as unpublished information. Of all 657, 55 are classified as unpublished information, most of which are as old as 1996. How reliable can be a piece of information supplied to the authors 25 years ago and never published since then?

*Indeed, old and unpublished information is the least reliable source. Unfortunately, those cases cannot be considered outdated sensu stricto due to the absence of more relevant information. We are grateful that the referee highlighted this topic, but cannot agree that it is a scientific weakness of the database; instead, it displays a bias in active fault studies towards most active or easily accessible fault systems.*

*Referee's concerns on the reliability have already been accounted for in the CONF (level of confidence) parameter.*

Unfortunately, the lowest confidence value "D" in the CONF field of the shapefile mixes up both published and unpublished materials. Also, many items classified as CONF=D are dissolved in regions where updated studies are available. So this reviewer confirms the scientific weakness, and the unclear communication to the users confirms the technical weakness.

In the last decade, several active fault databases have been published containing updated information. Below I list some of them (not necessarily exhaustively) that have significant geographical overlap with AFEAD and contain more up-to-date data than AFEAD.

- Europe (Atanackov et al., 2021; Caputo & Pavlides, 2013; DISS Working Group, 2018; European Geological Data Infrastructure, 2021; Ganas, 2021; Jomard et al., 2017; Vanneste et al., 2013)
- Middle East (Danciu et al., 2018)
- Central Asia (Mohadjer et al., 2016)
- Georgia (Onur et al., 2019, 2020)
- Japan (National Institute of Advanced Industrial Science and Technology, 2012)
- Africa (Williams et al., 2021)
- World (Christophersen et al., 2015; Styron & Pagani, 2020)

*Provided data will be included in the forthcoming update of the AFEAD v.2022; a portion of data has been already populated after the AFEAD v.2021 release. However, they are not comparable to AFEAD by extent or detail or both.*

The authors seem more concerned to reply to this reviewer than to inform their readers and potential database users about their intentions to update AFEAD or about the existence of these more up-to-date datasets.

Apart from those compilations released in the last year, most of these have been around for quite a long time now. In addition to this lack of data, the relationship between the fault representation in AFEAD and the fault representation in the source dataset is not clear. This is of particular concern for the blind faults since only criteria associated with the topographic signature are recalled. On the one hand, not considering the latest fault compilations prevents AFEAD from listing the newly recognized active faults. On the other hand, it also prevents AFEAD from eliminating those faults that were once considered active but are currently considered not active based on new evidence. Unfortunately, the CONF parameter does not consider the recency of the information.

*A workflow of transferring source data to the AFEAD representation is presented in section 4. Source Data. We have expanded this section to clarify the workflow, especially in the cases of contradiction among data sources. There is no direct relation between the recency of the information and its accuracy, so any join of recent data requires a comparison of the reasoning behind older and recent objects. The result of the comparison affects CONF in either its elevation or decrease and even deletion from the database.*

The added explanations sound more like an excuse not to add references to the most recent and likely more accurate works on active faulting than AFEAD. That there can be a time lag between the appearance of a publication and its ingestion into a database is perfectly understandable even without

saying. Some of the data products mentioned by this reviewer are over ten years old already, and not only did the authors not consider those data for inclusion in AFEAD, but they also neglected them in their discussion.

The compilation of the fault parameters also remains rather obscure in several aspects. For example, of the 47,363 faults, 22,270 (47%) have no parameter assigned (field "Parm" is NULL). Of the 25,093 faults with the field "Parm" not NULL, only 6,849 reports a "Rate=" value; how was then the Rate (rank) parameter assigned to the remaining faults?

*Objects of null "Parm" are typically those collected from fault maps with no parameterization. Please note that RATE=3 means "no measured rate above 1 mm/yr" (see Table 2), so it addresses all those cases.*

This reply does not clarify the issue. Firstly, there are 542 records with "Parm" = NULL and Rate < 3. Secondly, the definition of Rate=3 does not distinguish between "no measures at all" and "measures below 1 mm/yr but above 0 mm/year."

**Technical weaknesses**

The AFEAD is distributed as a single shapefile. Technically speaking, it is not even a database apart from the implicit relation between geographic features and their attributes. No relational table is provided between AFEAD and any of its linked information. In other words, it should be classified as a geographical flat-file, not a proper database.

*According to Wikipedia, "A database is an organized collection of data, generally stored and accessed electronically from a computer system." (https://en.wikipedia.org/wiki/Database), and AFEAD satisfies this definition of a database. However, it may not meet the definition of a relation database. Depending on the editor's decision, we can identify AFEAD as a "dataset" as it affects neither its inner structure nor representation. However, our experience in hosting and distribution of tectonic data shows that user-friendly shapefile format gets better reception among the researchers. Most AFEAD use cases require basic spatial analysis and text search on the user device without DBMS software.*

The authors retained only the first few words of the definition given by Wikipedia (https://en.wikipedia.org/wiki/Database). AFEAD has some linked information in a separate table which is not properly related to the main table.

The fields in the shapefile attribute table are very poorly organized.

First of all, none of the fields can be identified as a primary key. The lack of a primary key prevents the user from uniquely identifying any records and establishing their possible relations with external information. Also, the user cannot make an explicit reference to an individual AFEAD record when using it, including this review.

*A primary key has been added (field "FID").*

The FID field does not appear in the linked shapefiles (https://doi.org/10.13140/RG.2.2.10333.74726 last access on 26/02/2022).

Both the "Auth" and "Parm" fields contain long text strings that, in the next update, could become even longer and easily exceed the limitations imposed by the shapefile format. Notice that the maximum number of characters in a text field of a shapefile is 254, see Attribute limitations in ESRI documentation at: https://desktop.arcgis.com/en/arcmap/latest/manage-data/shapefiles/geoprocessing-considerations-for-shapefile-output.htm#GUID-A10ADA3B-0988-4AB1-9EBA-AD704F77B4A2

or

https://support.esri.com/en/technical-article/000012081

*Even accounting for shapefile standard limitations, we consider it the best format to distribute among researchers in the field of active faulting. It requires no proprietary software but supports spatial analysis and data queries. Only few objects are close to the maximum string length in AUTH or PARM and this could easily be resolved by removal of outdated or least relevant sources. In the current AFEAD schema, field limitations do not affect data presentation and usability.*

It is not the choice of the shapefile questioned but its use.

These two fields are also very difficult to explore, especially the Parm field that contains very heterogeneous parameters. This poor organization makes it hard for the user to use the database. For example, selecting the faults that have a certain "depth" information would require a very complex query, which would discourage the non-experts in SQL and expose the users to uncertain results. Also, the Parm field takes up more bytes than needed by repeating within the field the word to identify the parameter type, such as "Sense=" or "Rate=" or "Depth=", occasionally also including the reference to the parameter itself.

*Indeed, PARM is designated for ease of reading, not querying. Below, the reviewer proposes to "separate the "Parm" attributes into different columns, paying attention to storing single numerical values in individual columns." A schema of the spatial database of the World Map of Major Active Faults (DB96) was exactly what the reviewer suggest, and we intentionally changed this approach in AFEAD. The suggested schema leaves no room for different estimates of the same parameter and references for these estimations. A defined domain of values will distort citing of data (e.g. single numerical value is required where only value range or upper estimate is known). Finally, well above 90% of such fields will be empty, which hampers visual interaction with data. However, if any parameter, e.g. depth, becomes credible for a large amount of data, it will be recorded to an individual column (say, DEPTH), like it was done for fault sense (fields SENS1, SENS2) and uplifted side (field SIDE).*

The first statement in this reply contradicts the Database definition the authors proposed to adopt by referring to the Wikipedia definition. As for the ease of reading, do the authors think it is easy to scroll up and down a table with over 47 thousand entries for locating those with some parametric characterization?

It's a shame, however, to learn the authors already had such a more appropriate database schema and downgraded their work to this confusing and inefficient design of AFEAD. The schema suggested by this reviewer requires only some data manipulation and reorganization that would improve the AFEAD usability. A properly designed database including one-to-many relational tables would solve the issue regarding the multiple interpretations.

The use of the "+" (plus) sign in the "Side" field is unnecessary because all the non-null values are a plus. It could also be troublesome because the plus sign can be automatically converted when importing the data in other systems (try saving the attribute table into the Microsoft Excel format, for example).

*SIDE is a text field, and any DBMS may handle mathematical symbols in text strings. We were unable to reproduce problems when opening .dbf attribute table in MS Excel. In active faults databases, it is common to label a downthrown side as well, so the plus sign serves as a reminder about an uplifted side.*

Open AFEAD shapefile in QGIS, save the layer as CSV, open AFEAD.csv in MS Excel and see all the values in column "E" (SIDE) showing the Excel message "#NAME?" with the content of the cells reading "=+W" since the "+" sign is interpreted as being part of a formula.

[Figure]

| | A | B | C | D | E | F | G | H | I | J | K | L | M |
|---|---|---|---|---|---|---|---|---|---|---|---|---|---|
| 1 | RATE | CONF | SENS1 | SENS2 | SIDE | TEXT | FAULT_NA | ZONE_NA | AUTH | PARM | | | |
| 23 | 3 | B | E | | | (?) Rift valley | | | Bird, 2003 | | | | |
| 24 | 2 | B | R | | #NAME? | Underwat | Owen Fra | Owen Fra | McKenzie | Signs= EQ \| Rate= D 16; Seism= M5.0-5.9; Se | | | |
| 25 | 1 | A | R | | #NAME? | Sharp und | Oman | | Makran Tr | Map of Mi | Signs= EQ \| Side= (?) +S (DB, 1996) \| Sense= | | |
| 26 | 3 | D | E | | | Underwat | Bab El Ma | Red Sea R | Working n | Signs= EQ \| Seism= M5.0 2005-04-30 D11 kn | | | |
| 27 | 1 | B | E | | | Rift valley | | Tajura Rift | McKenzie et al., 1971; Kanaev et al., 1976, 1977; Map o | | | |
| 28 | 3 | B | U | | | Underwat | Shukra El | Shiek | | Kanaev et al., 1977; Map of Middle East, 1992; Lukina, | | | |
| 29 | 1 | A | E | | | Rift valley | | West Shel | McKenzie | Signs= EQ \| Seism= M5.6 2009-11-05 D10 kn | | | |
| 30 | 3 | B | U | | | Underwater valley \| Eqs M<5, N | Lisitsyn et | Signs= EQ \| Rate=D15-20 (DB, 1996) | | | | |
| 31 | 3 | B | N | | #NAME? | Underwater hill esca | Tajura Rift | Lisitsyn et | Signs= EQ \| Rate= E15-20; Seism= M<5; M5- | | | |
| 32 | 2 | B | N | | #NAME? | Hill escarpment \| Co | Tajura Rift | Lisitsyn et | Signs= EQ \| Rate= 15-20; Seism= M5-6 (DB, | | | |
| 33 | 3 | B | U | | | Underwater valley | | Lisitsyn et | Signs= EQ \| Rate=D15-20; Seism= M<5 (DB, | | | |
| 34 | 3 | B | N | | #NAME? | Underwater hill esca | Tajura Rift | Lisitsyn et | Rate= E15-20; Seism= M5 (DB, 1996) | | | | |
| 35 | 2 | A | N | | #NAME? | Hill escarpment \| Co | Tajura Rift | Lisitsyn et | Signs= EQ \| Seism= M<5 \| Sense= E, N; Rate | | | |
| 36 | 3 | B | U | | | Underwater valley | | Lisitsyn et | Rate= D15-20 (DB, 1996) | | | | |
| 37 | 3 | B | N | | #NAME? | Underwater hill esca | Tajura Rift | Lisitsyn et | Rate= E15-20 (Bird, 2003) | | | | |
| 38 | 2 | B | N | | #NAME? | Hill escarpment \| Co | Tajura Rift | Lisitsyn et | Signs= EQ \| Sense= E, N; Rate= 15-20 (Bird, | | | |
| 39 | 3 | B | U | | | Underwater valley | | Lisitsyn et | Rate= D15-20 (DB, 1996) | | | | |
| 40 | 3 | B | N | | #NAME? | Underwater hill esca | Tajura Rift | Lisitsyn et | Seism= M5 (DB, 1996) | | | | |
| 41 | 3 | B | U | | | Underwater valley \| Eqs M<5 (D | Lisitsyn et | Signs= EQ \| Rate= D15-20 (DB, 1996) | | | | |
| 42 | 3 | B | N | | #NAME? | Underwater hill esca | Tajura Rift | Lisitsyn et | Rate= E15-20 (Bird, 2003) | | | | |
| 43 | 2 | B | N | | #NAME? | Underwater hill esca | Tajura Rift | Lisitsyn et | Signs= EQ \| Rate= E15-20 (Bird, 2003) | | | |
| 44 | 3 | B | U | | | Underwater valley | | Lisitsyn et | Rate= D15-20 (DB, 1996) | | | | |
| 45 | 3 | B | N | | #NAME? | Underwater hill esca | Tajura Rift | Lisitsyn et | Rate= E15-20 (Bird, 2003) | | | | |
| 46 | 2 | B | N | | #NAME? | Underwater hill esca | Tajura Rift | Lisitsyn et | Signs= EQ \| Seism= Epicenters, M-unknow; | | | |
| 47 | 3 | B | U | | | Underwater valley | | Lisitsyn et | Rate= D15-20; Seism= M5-6 (DB, 1996) | | | | |
| 48 | 3 | C | N | | #NAME? | Underwater hill esca | Tajura Rift | Lisitsyn et | Seism= M5-6; Rate= E15-20 (DB, 1996) | | | |
| 49 | 2 | A | N | | #NAME? | Underwater hill esca | Tajura Rift | Lisitsyn et | Signs= EQ \| Rate= E15-20 (Bird, 2003) \| Seis | | |

Simply renaming column SIDE as UPSIDE and getting rid of the "+" would have solved the problem.

**Other issues**

L1: The name of the database does not reflect its abbreviation "AFEAD" should be "Active Faults of Eurasia Database," not "Database of the Active Faults of Eurasia." Please make a choice and stick to it.

L14: In the file provided, the sources are 657, not 612. The difference is 55, which corresponds to the number of unpublished work. Rephrase to make this clear for the readers.

L25: Unclear reference to "Geologische Rundschau, 1955"; see also L327.

L166: Unclear to whom "our team" is referring.

L72-74: This statement is unclear, or it is at least quite questionable. Linear landforms created by nontectonic processes are not rare, and several earthquakes have reactivated faults with very complex patterns. Also, cases of tectonic inversion are known. Maybe the authors can expand this paragraph to make it clearer and more documented for what they want to say.

L91-92: Is the fold axis represented? Otherwise, which element of the structure is represented? And how can the user be aware of that?

Table 1: Is the strike-slip with unknown sense contemplated?

*They suggest straightforward corrections to the manuscript, all of them were accepted. In AFEAD, strike-slip with unknown sense is considered equal to unknown sense (SENS1=U).*

The SENS1 definition remains unclear and insufficient. SENS1=V and SENS1=U are not mutually exclusive.

**Recommendations**

The following few technical fixes are necessary to make AFEAD suitable for using it in a proper DBMS.

*We consider shapefile to be the most suitable data format for the distribution of AFEAD at the moment. The provided guidelines will be essential for a redesign of AFEAD when demand for relation database managed by DBMS software increases.*

The problem raised by this reviewer has nothing to do with the shapefile format.

- Establish a primary key that uniquely identifies each record (fault) of the shapefile.
- Separate the "Parm" attributes into different columns, paying attention to storing single numerical values in individual columns.
- Establish a primary key for the table of bibliographic references.
- Create a relational table (many to many) that connects the fault table primary keys with the bibliographic reference table primary keys.
- Once the relational table is created, the column "Auth" can be deleted from the shapefile.
- Remove all "+" "-" "=" and similar signs/symbols from all columns. Use the "+" or "-" sign only with numerical values.

The European plate boundary along the Mid-Atlantic Ridge should be completed to make AFEAD adhere to its name (it could be disappointing for the AFEAD user to find data in the African plate and not the complete European plate).

*Faults in the Mid-Atlantic Ridge will be included in the forthcoming update of the AFEAD v.2022*

Again, the authors should inform their potential readers, not just the reviewer.

More explanations are needed to make the user understand the source of information used to assign the Rate ranks.

*Explanations have been added to the manuscript and AFEAD web map interface.*

What was added is not a sufficient explanation for the AFEAD user to understand the source of information.

A justification is needed for not considering all the recent fault data compilations published in the last decade. The authors should also discuss the implications due to the lack of updated information and warn the users about the limitations in using AFEAD instead of more up-to-date regional/local data.

*Explanations have been added to the manuscript and AFEAD web map interface.*

Repeated from above: The added explanations sound more like an excuse not to add references to most recent, and very likely more accurate works on active faulting than AFEAD. That there can be a time lag between the appearance of a publication and its ingestion into a database is perfectly understandable even without saying. Some of the data products mentioned by this reviewer are over ten years old already, and not only the authors did not consider those data for inclusion in AFEAD but they also neglected them in their discussion.

**References**

Atanackov, J., Jamšek Rupnik, P., Jež, J., Celarc, B., Novak, M., Milanič, B., et al. (2021). Database of Active Faults in Slovenia: Compiling a New Active Fault Database at the Junction Between the Alps, the Dinarides and the Pannonian Basin Tectonic Domains. *Frontiers in Earth Science*, *9*, 604388. https://doi.org/10.3389/feart.2021.604388

Caputo, R., & Pavlides, S. (2013). Greek Database of Seismogenic Sources (GreDaSS): A compilation of potential seismogenic sources (Mw > 5.5) in the Aegean Region [Text/html,application/vnd.google-earth.kml+xml,image/jpg]. University of Ferrara, Italy. https://doi.org/10.15160/UNIFE/GREDASS/0200

Christophersen, A., Litchfield, N., Berryman, K., Thomas, R., Basili, R., Wallace, L., et al. (2015). Development of the Global Earthquake Model's neotectonic fault database. *Natural Hazards*, *79*(1), 111–135. https://doi.org/10.1007/s11069-015-1831-6

Danciu, L., Şeşetyan, K., Demircioglu, M., Gülen, L., Zare, M., Basili, R., et al. (2018). The 2014 Earthquake Model of the Middle East: seismogenic sources. *Bulletin of Earthquake Engineering*, *16*(8), 3465–3496. https://doi.org/10.1007/s10518-017-0096-8

DISS Working Group. (2018, April 26). Database of Individual Seismogenic Sources (DISS), version 3.2.1. Istituto Nazionale di Geofisica e Vulcanologia (INGV), DOI: 10.6092/INGV.IT-DISS3.2.1. Retrieved July 14, 2020, from http://diss.rm.ingv.it/diss/

European Geological Data Infrastructure. (2021). HIKE European Fault Database. Retrieved from

    https://geoera.eu/projects/hike10/faultdatabase/

Ganas, A. (2021). NOAFAULTS KMZ layer Version 3.0.1 (2021 update) (Version V3.0.1) [Data set].

    Zenodo. https://doi.org/10.5281/ZENODO.4897894

Jomard, H., Cushing, E. M., Palumbo, L., Baize, S., David, C., & Chartier, T. (2017). Transposing an active

    fault database into a seismic hazard fault model for nuclear facilities – Part 1: Building a

    database of potentially active faults (BDFA) for metropolitan France. *Natural Hazards and Earth*

    *System Sciences*, *17*(9), 1573–1584. https://doi.org/10.5194/nhess-17-1573-2017

Mohadjer, S., Ehlers, T. A., Bendick, R., Stübner, K., & Strube, T. (2016). A Quaternary fault database for

    central Asia. *Natural Hazards and Earth System Sciences*, *16*(2), 529–542.

    https://doi.org/10.5194/nhess-16-529-2016

National Institute of Advanced Industrial Science and Technology. (2012). Active Fault Database of

    Japan, February 28, 2012 version. Research Information Database DB095, National Institute of

    Advanced Industrial Science and Technology. Retrieved from

    https://gbank.gsj.jp/activefault/index_e_gmap.html

Onur, T., Gok, R., Godoladze, T., Gunia, I., Boichenko, G., Buzaladze, A., et al. (2019). *Probabilistic Seismic*

    *Hazard Assessment for Georgia* (No. LLNL-TR--771451, 1511856) (p. LLNL-TR--771451, 1511856).

    https://doi.org/10.2172/1511856

Onur, T., Gok, R., Godoladze, T., Gunia, I., Boichenko, G., Buzaladze, A., et al. (2020). Probabilistic Seismic

    Hazard Assessment Using Legacy Data in Georgia. *Seismological Research Letters*, *91*(3), 1500–

    1517. https://doi.org/10.1785/0220190331

Styron, R., & Pagani, M. (2020). The GEM Global Active Faults Database. *Earthquake Spectra*,

    *36*(1_suppl), 160–180. https://doi.org/10.1177/8755293020944182

Vanneste, K., Camelbeeck, T., & Verbeeck, K. (2013). A Model of Composite Seismic Sources for the

Lower Rhine Graben, Northwest Europe. *Bulletin of the Seismological Society of America*,

*103*(2A), 984–1007. https://doi.org/10.1785/0120120037

Williams, J. N., Wedmore, L. N. J., Scholz, C. A., Kolawole, F., Wright, L. J. M., Shillington, D., et al. (2021).

*The Malawi Active Fault Database: an onshore-offshore database for regional assessment of*

*seismic hazard and tectonic evolution* (preprint). Geophysics.

https://doi.org/10.1002/essoar.10507158.1

---

## Author Response (AR2)

**Response to referee's second review of "The Active Faults of Eurasia Database (AFEAD): Ontology and Design behind the Continental-Scale Dataset" submitted by Egor Zelenin et al.**

We are deeply grateful to the reviewer for their careful examination of the manuscript and AFEAD itself. The authors carefully considered the raised issues and regret that the provided justification was considered insufficient. We recognize these reviews as a strong reason for major revisions of the manuscript, and we have made such a revision. Among the main improvements of the manuscript are paragraphs considering other active fault databases, an expanded description of attributes, and a new section on the strategy of database update. In addition to the corrected manuscript and database, we provide the following point-to-point answer (reviewer's comments in *italic, red is the text of the second review,* and blue are answers to it).

**Scientific weaknesses**

1. *The data collection is based on bibliographical investigations, but most of the bibliographic references are quite outdated. Out of the 657 references (in the Excel file), only 13 are post-2010. Of these 13, three are classified as unpublished information. Of all 657, 55 are classified as unpublished information, most of which are as old as 1996. How reliable can be a piece of information supplied to the authors 25 years ago and never published since then?*

   Indeed, old and unpublished information is the least reliable source. Unfortunately, those cases cannot be considered outdated *sensu stricto* due to the absence of more relevant information. We are grateful that the referee highlighted this topic, but cannot agree that it is a scientific weakness of the database; instead, it displays a bias in active fault studies towards most active or easily accessible fault systems. Referee's concerns on the reliability have already been accounted for in the CONF (level of confidence) parameter.

   *Unfortunately, the lowest confidence value "D" in the CONF field of the shapefile mixes up both published and unpublished materials. Also, many items classified as CONF=D are dissolved in regions where updated studies are available. So this reviewer confirms the scientific weakness, and the unclear communication to the users confirms the technical weakness.*

   We are constantly working on an update of AFEAD. Since the initial submission, data from 19 recent (2013-2021) studies have been collected. However, the update of the database does not imply replacing all the preceding data within the spatial extent of the update. Collisions between AFEAD and new data are resolved via adjustment of CONF values (see section Update Strategy). The separate metric of reliability is required as the age of publication and even the fact of publication does not directly affect the reliability of data. Objects within the extent of updated studies may remain in AFEAD even if absent in these recent studies, and CONF allows us to manage the likelihood of activity. Of the 55 unpublished studies mentioned by the reviewer, 50 are those compiled for the World Map of Major Active Faults (see sections 3.4 Reference List and 4 Source Data). Their spatial location has been published (Trifonov, 1997, 2004), but with fewer attributes than in AFEAD. The World Map of Major Active Faults was a major compilation of field data and continuous mapping, and indeed, some faults have never been studied since then due to their remoteness or low expected hazard.

2. *In the last decade, several active fault databases have been published containing updated information. Below I list some of them (not necessarily exhaustively) that have significant geographical overlap with AFEAD and contain more up-to-date data than AFEAD.*

Provided data will be included in the forthcoming update of the AFEAD v.2022; a portion of data has been already populated after the AFEAD v.2021 release. However, they are not comparable to AFEAD by extent or detail or both.

*The authors seem more concerned to reply to this reviewer than to inform their readers and potential database users about their intentions to update AFEAD or about the existence of these more up-to-date datasets.*

We have added a paragraph considering recent active fault databases in the Introduction (lines 44-49). Iterative update of AFEAD was announced in the former section Data Access and Further Development and now is expanded in section 6 Update Strategy.

3. *Apart from those compilations released in the last year, most of these have been around for quite a long time now. In addition to this lack of data, the relationship between the fault representation in AFEAD and the fault representation in the source dataset is not clear. This is of particular concern for the blind faults since only criteria associated with the topographic signature are recalled. On the one hand, not considering the latest fault compilations prevents AFEAD from listing the newly recognized active faults. On the other hand, it also prevents AFEAD from eliminating those faults that were once considered active but are currently considered not active based on new evidence. Unfortunately, the CONF parameter does not consider the recency of the information.*

A workflow of transferring source data to the AFEAD representation is presented in section 4. Source Data. We have expanded this section to clarify the workflow, especially in the cases of contradiction among data sources. There is no direct relation between the recency of the information and its accuracy, so any join of recent data requires a comparison of the reasoning behind older and recent objects. The result of the comparison affects CONF in either its elevation or decrease and even deletion from the database.

*The added explanations sound more like an excuse not to add references to the most recent and likely more accurate works on active faulting than AFEAD. That there can be a time lag between the appearance of a publication and its ingestion into a database is perfectly understandable even without saying. Some of the data products mentioned by this reviewer are over ten years old already, and not only did the authors not consider those data for inclusion in AFEAD, but they also neglected them in their discussion.*

All the provided references have been thoroughly considered and included in the AFEAD, as well as some recent studies not mentioned by the reviewer. We have provided additional information on the workflow of transferring source data to AFEAD in section 6 Update Strategy.

4. *The compilation of the fault parameters also remains rather obscure in several aspects. For example, of the 47,363 faults, 22,270 (47%) have no parameter assigned (field "Parm" is NULL). Of the 25,093 faults with the field "Parm" not NULL, only 6,849 reports a "Rate=" value; how was then the Rate (rank) parameter assigned to the remaining faults?*

Objects of null "Parm" are typically those collected from fault maps with no parameterization. Please note that RATE=3 means "no measured rate above 1 mm/yr" (see Table 2), so it addresses all those cases.

*This reply does not clarify the issue. Firstly, there are 542 records with "Parm" = NULL and Rate < 3. Secondly, the definition of Rate=3 does not distinguish between "no measures at all" and "measures below 1 mm/yr but above 0 mm/year."*

Populating of derivative attributes, including RATE, has been described in the expanded section 3.3 Derivative Attributes (lines 166-169) and the new section 6 Update Strategy.

**Technical weaknesses**

5. *The AFEAD is distributed as a single shapefile. Technically speaking, it is not even a database apart from the implicit relation between geographic features and their attributes. No relational table is provided between AFEAD and any of its linked information. In other words, it should be classified as a geographical flat-file, not a proper database.*

According to Wikipedia, "A database is an organized collection of data, generally stored and accessed electronically from a computer system." (https://en.wikipedia.org/wiki/Database), and AFEAD satisfies this definition of a database. However, it may not meet the definition of a relation database. Depending on the editor's decision, we can identify AFEAD as a "dataset" as it affects neither its inner structure nor representation. However, our experience in hosting and distribution of tectonic data shows that user-friendly shapefile format gets better reception among the researchers. Most AFEAD use cases require basic spatial analysis and text search on the user device without DBMS software.

*The authors retained only the first few words of the definition given by Wikipedia (https://en.wikipedia.org/wiki/Database). AFEAD has some linked information in a separate table which is not properly related to the main table.*

Considering the policy of the Earth System Science Data journal, of 16 geology-related databases, not datasets, published in ESSD since 2017, only two are relation databases published in SQL format. The remaining 14 are flat-files, excel tables, or shapefiles, still named "databases". Judging by these cases as well as our understanding of the broad "database" term, we consider "database" acceptable. However, we are ready to refer to AFEAD as a "dataset," rather than a "database," based on the editor's decision and to make any necessary changes to the manuscript.

*The fields in the shapefile attribute table are very poorly organized. First of all, none of the fields can be identified as a primary key. The lack of a primary key prevents the user from uniquely identifying any records and establishing their possible relations with external information. Also, the user cannot make an explicit reference to an individual AFEAD record when using it, including this review.*

A primary key has been added (field "FID").

*The FID field does not appear in the linked shapefiles (https://doi.org/10.13140/RG.2.2.10333.74726 last access on 26/02/2022).*

Unique Fault ID has been added explicitly (https://doi.org/10.13140/RG.2.2.32655.05280).

6. *Both the "Auth" and "Parm" fields contain long text strings that, in the next update, could become even longer and easily exceed the limitations imposed by the shapefile format. Notice that the maximum number of characters in a text field of a shapefile is 254, see Attribute limitations in ESRI documentation at: https://desktop.arcgis.com/en/arcmap/latest/manage-data/shapefiles/geoprocessingconsiderations-for-shapefile-output.htm#GUID-A10ADA3B-0988-4AB1-9EBA-AD704F77B4A2 or https://support.esri.com/en/technical-article/000012081*

Even accounting for shapefile standard limitations, we consider it the best format to distribute among researchers in the field of active faulting. It requires no proprietary software but supports spatial analysis and data queries. Only few objects are close to the maximum string length in AUTH or PARM and this could easily be resolved by removal of outdated or least relevant sources. In the current AFEAD schema, field limitations do not affect data presentation and usability.

*It is not the choice of the shapefile questioned but its use.*

Reasoning behind the questioned approach has been described in both expanded section 3.2 (lines 116-120) and new section 6 Update Strategy.

7. *These two fields are also very difficult to explore, especially the Parm field that contains very heterogeneous parameters. This poor organization makes it hard for the user to use the database. For example, selecting the faults that have a certain "depth" information would require a very complex query, which would discourage the non-experts in SQL and expose the users to uncertain results. Also, the Parm field takes up more bytes than needed by repeating within the field the word to identify the parameter type, such as "Sense=" or "Rate=" or "Depth=", occasionally also including the reference to the parameter itself.*

Indeed, PARM is designated for ease of reading, not querying. Below, the reviewer proposes to "separate the "Parm" attributes into different columns, paying attention to storing single numerical values in individual columns." A schema of the spatial database of the World Map of Major Active Faults (DB96) was exactly what the reviewer suggest, and we intentionally changed this approach in AFEAD. The suggested schema leaves no room for different estimates of the same parameter and references for these estimations. A defined domain of values will distort citing of data (e.g. single numerical value is required where only value range or upper estimate is known). Finally, well above 90% of such fields will be empty, which hampers visual interaction with data. However, if any parameter, e.g. depth, becomes credible for a large amount of data, it will be recorded to an individual column (say, DEPTH), like it was done for fault sense (fields SENS1, SENS2) and uplifted side (field SIDE).

*The first statement in this reply contradicts the Database definition the authors proposed to adopt by referring to the Wikipedia definition. As for the ease of reading, do the authors think it is easy to scroll up and down a table with over 47 thousand entries for locating those with some parametric characterization? It's a shame, however, to learn the authors already had such a more appropriate database schema and downgraded their work to this confusing and inefficient design of AFEAD. The schema suggested by this reviewer requires only some data manipulation and reorganization that would improve the AFEAD usability. A properly designed database including one-to-many relational tables would solve the issue regarding the multiple interpretations.*

In addition to the previous answer, the current database design is a trade-off between the ease of computer processing and user accessibility. The former "separate attributes" design of DB96, even seeming more appropriate, proved ineffective due to the excessive number of null values and the intention to record different estimates of the same parameter. As for the previous issue, the reasoning behind the questioned approach has been described in both expanded section 3.2 (lines 116-121, 135-138) and the new section 6 Update Strategy.

8. *The use of the "+" (plus) sign in the "Side" field is unnecessary because all the non-null values are a plus. It could also be troublesome because the plus sign can be automatically converted when importing the data in other systems (try saving the attribute table into the Microsoft Excel format, for example).*

SIDE is a text field, and any DBMS may handle mathematical symbols in text strings. We were unable to reproduce problems when opening .dbf attribute table in MS Excel. In active faults databases, it is common to label a downthrown side as well, so the plus sign serves as a reminder about an uplifted side.

*Open AFEAD shapefile in QGIS, save the layer as CSV, open AFEAD.csv in MS Excel and see all the values in column "E" (SIDE) showing the Excel message "#NAME?" with the content of the cells reading "=+W" since the "+" sign is interpreted as being part of a formula. Simply renaming column SIDE as UPSIDE and getting rid of the "+" would have solved the problem.*

Converted to UPSIDE, "+" removed.

**Other issues (omitted are those solved after the first review)**

9. *Table 1: Is the strike-slip with unknown sense contemplated?*

In AFEAD, strike slip with unknown sense is considered equal to unknown sense (SENS1=U).

*The SENS1 definition remains unclear and insufficient. SENS1=V and SENS1=U are not mutually exclusive.*

SENS1 definition expanded (Table 1 and lines 170-173).

**Recommendations**

10. *The following few technical fixes are necessary to make AFEAD suitable for using it in a proper DBMS.*

We consider shapefile to be the most suitable data format for the distribution of AFEAD at the moment. The provided guidelines will be essential for a redesign of AFEAD when demand for relation database managed by DBMS software increases.

*The problem raised by this reviewer has nothing to do with the shapefile format.*

Each technical advice is answered below:

- *Establish a primary key that uniquely identifies each record (fault) of the shapefile.*

Unique fault ID has been added.

- *Separate the "Parm" attributes into different columns, paying attention to storing single numerical values in individual columns.*

Following the described strategy, some parameters are likely to be separated in the future, as it was done for RATE or SENS1. However, it will not affect "PARM" due to concerns described in section 3 Database Model and the new section 6 Update Strategy.

- *Establish a primary key for the table of bibliographic references.*

There is a primary key – a "citation" column.

- *Create a relational table (many to many) that connects the fault table primary keys with the bibliographic reference table primary keys.*

The relation table is provided.

- *Once the relational table is created, the column "Auth" can be deleted from the shapefile.*

Deletion of AUTH will not improve usability and may impede the most common use case of textual analysis of spatial query to AFEAD by an inexperienced user.

- *Remove all "+" "-" "=" and similar signs/symbols from all columns. Use the "+" or "-" sign only with numerical values.*

Only leading symbols may affect representation of data in external software, and those have been removed.

11. *The European plate boundary along the Mid-Atlantic Ridge should be completed to make AFEAD adhere to its name (it could be disappointing for the AFEAD user to find data in the African plate and not the complete European plate).*

Faults in the Mid-Atlantic Ridge will be included in the forthcoming update of the AFEAD v.2022

*Again, the authors should inform their potential readers, not just the reviewer.*

Faults in the Mid-Atlantic Ridge have been included. Potential readers have been informed about the spatial domain of the AFEAD.

12. *More explanations are needed to make the user understand the source of information used to assign the Rate ranks.*

Explanations have been added to the manuscript and AFEAD web map interface.

*What was added is not a sufficient explanation for the AFEAD user to understand the source of information.*

More explanations have been added to the section 3 Database Model, an example of data processing, including rate assignment, have been added to the section 6 Update Strategy

13. *A justification is needed for not considering all the recent fault data compilations published in the last decade. The authors should also discuss the implications due to the lack of updated information and warn the users about the limitations in using AFEAD instead of more up-to-date regional/local data.*

Explanations have been added to the manuscript and AFEAD web map interface.

*Repeated from above: The added explanations sound more like an excuse not to add references to most recent, and very likely more accurate works on active faulting than AFEAD. That there can be a time lag between the appearance of a publication and its ingestion into a database is perfectly understandable even without saying. Some of the data products mentioned by this reviewer are over ten years old already, and not only the authors did not consider those data for inclusion in AFEAD but they also neglected them in their discussion.*

All the provided references have been included in the AFEAD, as well as other recent papers. We have added a notification about lacking publications to the Introduction according to the reviewer advice (lines 53-54).